# Contemporary Strategies for the Structural Design of Multi-Story Modular Timber Buildings: A Comprehensive Review

Marina Tenório [1,*], Rui Ferreira [2], Victor Belafonte [1], Filipe Sousa [2], Cláudio Meireis [2], Mafalda Fontes [2], Inês Vale [1], André Gomes [1], Rita Alves [2], Sandra M. Silva [1], Dinis Leitão [1], André Fontes [2], Carlos Maia [2], Aires Camões [1] and Jorge M. Branco [1]

[1] Department of Civil Engineering, Institute for Sustainability and Innovation in Engineering Structures (ISISE), University of Minho, Campus de Azurém, Av. da Universidade, 4800-058 Guimarães, Portugal; victorbelafonte@gmail.com (V.B.); ines.pvale@gmail.com (I.V.); andrelg1810@gmail.com (A.G.); sms@civil.uminho.pt (S.M.S.); dleitao@civil.uminho.pt (D.L.); aires@civil.uminho.pt (A.C.); jbranco@civil.uminho.pt (J.M.B.)

[2] Lab2PT, School of Architecture, Art and Design, University of Minho, Campus de Azurém, Av. da Universidade, 4800-058 Guimarães, Portugal; rui_ferreira_3@hotmail.com (R.F.); filipe_sousas@hotmail.com (F.S.); claudio13dias@gmail.com (C.M.); mafaldaflcfontes@gmail.com (M.F.); ritaalves.arq@gmail.com (R.A.); afontes@cfaarch.com (A.F.); ocarlosmaia@gmail.com (C.M.)

* Correspondence: tenorio.mcu@gmail.com

**Abstract:** Modular timber construction embodies a pioneering and eco-friendly methodology within the building sector. With the notable progress made in manufacturing technologies and the advent of engineered wood products, timber has evolved into a promising substitute for conventional materials such as concrete, masonry, and steel. Beyond its structural attributes, timber brings environmental advantages, including its inherent capacity for carbon sequestration and a reduced carbon footprint compared to conventional materials. Timber's lightweight nature, coupled with its versatility and efficiency in factory-based production, accelerates modular construction processes, providing a sustainable solution to the growing demands of the building industry. This work thoroughly explores contemporary modular construction using wood as the primary material. The investigation spans various aspects, from the fundamentals of modularity and the classification of modular timber solutions to considerations of layout design, structural systems, and stability at both the building and module levels. Moreover, inter-module joining techniques, MEP (mechanical, electrical, and plumbing) integration, and designs for disassembly are scrutinized. The investigation led to the conclusion that timber modular construction, drawing inspiration from the steel modular concept, consistently utilizes a structural approach based on linear members (timber frame, post-and-beam, etc.), incorporating stability configurations and diverse joint techniques. Despite the emphasis on modularization and prefabrication for adaptability, a significant portion of solutions still concentrate on the on-site linear assembly process of those linear members. Regarding modularity trends, the initial prevalence of 2D and 3D systems has given way to a recent surge in the utilization of post-and-beam structures, congruent with the ascending verticality of buildings. In contrast to avant-garde and bold trends, timber structures typically manifest as rectilinear, symmetric plans, characterized by regular and repetitive extrusions, demonstrating a proclivity for centrally located cores. This work aims to offer valuable insights into the current utilization of modular timber construction while identifying pivotal gaps for exploration. The delineation of these unexplored areas seeks to enable the advancement of modular timber projects and systems, fully leveraging the benefits provided by prefabrication and modularity.

**Keywords:** timber buildings; modular construction; timber structures; prefabrication; modularity; joining techniques; MEP installations; disassembly

## 1. Introduction

The alarming global climate scenario, which has rapidly worsened in recent decades, has highlighted the urgency of reversing the consumption patterns practiced by various economic sectors, demanding the establishment of increasingly ambitious collective climate and energy mitigation goals. As a growth strategy for a modern, competitive, resource-efficient economy which is, consequently, conducive to sustainable development, one of the priorities outlined by the European Union is achieving carbon neutrality by 2050 [1].

In this context, the prioritization of revising the traditional models replicated by the construction sector becomes crucial when confronted with data regarding their impact. The production and processing of materials for the sector hold the most significant share of energy consumption and greenhouse gas emissions [2], with cement and steel accounting for 4 to 7% [3] and 5% [4] of global $CO_2$ emissions per year, respectively.

Wood, then, emerges as a promising construction material as an alternative to concrete and steel, with the benefit of its inherent contribution to emission reduction through its carbon storage capacity—approximately 0.9 t $CO_2/m^3$ of material—and lower embodied energy [5].

The population growth, coupled with the urbanization trend that will lead more than 2/3 of the global population to live in urban areas by 2050 [6], requires the establishment of increasingly dense urban forms, which often translates into high-rise buildings. According to the UK Committee on Climate Change [7], by 2060, 230 billion square meters of new construction will be added, with 2 billion square meters per year between 2019 and 2025 [8], contributing to the release of approximately 415 $GtCO_2$ into the atmosphere over the next 40 years [9]. Thus, population and urban expansion turn cities into opportunities for sustainable development [10], highlighting the potential of the timber industry to provide carbon-neutral solutions [11].

Constructing tall buildings with timber under the current standards is a pioneering effort made possible by innovations in the timber-based industry, both at the product and process levels. The introduction of Engineered Wood Products (EWP), such as laminated veneer lumber (LVL), and, in particular, cross-laminated timber (CLT), has optimized the structural properties of timber and overcome many of its limitations [12], reinventing and propelling the industry with products that exhibit high stability, rigidity, and competitive mechanical properties compared to concrete and steel [13].

On the other hand, the development of CAD/CAM (Computer-Aided Design/Computer-Aided Manufacturing) technology combined with the use of CNC (Computer Numeric Control) equipment and the integrated approach of various specialties within the BIM methodology has increased the precision and quality of wood products. Produced in a controlled factory environment, these products minimize on-site activity, thereby reducing construction time, site costs, noise, pollution, waste, accident risks, the need for traditional construction equipment, and material deterioration due to exposure to moisture [14–18]. An example is the Forté building, located in Melbourne and inaugurated in 2013. At that time, it was considered the largest timber apartment complex in the world, with 10 floors and 30% less construction time than that estimated for a similar structure using conventional materials [19]. According to APA [20], using a case study analysis, opting for prefabricated wood panels over conventional materials can result in even more substantial assembly time savings, exceeding 50%.

The objectives of this state-of-the-art review on timber modular construction are multifaceted, aiming to present a comprehensive and innovative analysis of the current landscape of this modern and promising construction typology that integrates sustainable building practices and efficient resource utilization. The first specific objective is to investigate and map current trends in timber modular construction, considering technological advancements, market changes, and recent innovations. This includes exploring production methods and design techniques that can positively influence the efficiency and sustainability of modular construction. This approach will enable us to gain an understanding of the contemporary scenario and identify knowledge gaps. Subsequently, the

review aims to identify the challenges faced by timber modular construction and examine solutions proposed in the recent literature. This will not only facilitate an understanding of existing limitations but also highlight innovative approaches to overcoming these challenges. Lastly, the review seeks to provide insights and future directions for research and practice in timber modular construction. This involves synthesizing our findings to formulate practical recommendations and suggestions for future investigations, aiming for continuous advancement and innovation in this dynamic and ever-evolving field.

## 2. Methodology

This state-of-the-art review employed a systematic approach to identify, analyze, and synthesize the relevant literature. The methodology involved the following steps:

- Scope definition: Clearly defined geographical, temporal, and thematic boundaries were established through a preliminary literature review, mapping the current landscape of modular timber construction, and identifying knowledge gaps. Documents published before the period of 1990–2023 were disregarded from the review;
- Source selection: Rigorous selection of information sources included academic databases, specialized journals, conferences, books, chapters, and technical reports, all written in English. The search engines used were Google Scholar and Google Books. These platforms provide a wide range of scholarly literature across disciplines, offering free accessibility and user-friendly interfaces for efficient navigation. Additionally, the inclusion of citation metrics assists in evaluating the impact of works, facilitating the identification of influential papers and authors. The literature search utilized specific keywords such as "timber", "wooden", "modular", "buildings", "construction", "structures", "systems", "prefabrication", "EWP", "modularity", "design", "guide", "layout", "multi-story", "module", "panel", "frame", "transport", "joining techniques", "connections", "inter-module", "MEP", "infrastructures", "installation", "stability", "performance", "disassembly", and "innovations". The keywords were consolidated using Boolean operators such as "and", indicating that all search terms must be present in the retrieved records, "or", indicating that any of the search terms can be present, and "not", instructing the database to disregard concepts that might be implicit in the search terms;
- Screening and classification: A screening process based on relevance and quality criteria was applied to the identified documents. Selected works were categorized thematically, facilitating an effective analysis of the diverse approaches and perspectives in the existing literature;
- Analysis and synthesis: A critical analysis of the selected articles identified trends, challenges, innovations, and gaps in current modular timber construction research. The information synthesis aimed to construct a cohesive narrative highlighting the present state of knowledge in the field, incorporating various perspectives and relevant findings;
- Article structuring: The article structure followed a logical sequence reflecting historical evolution, recent developments, and future projections for modular timber construction. Each section was carefully organized to provide a comprehensive and comprehensible overview of the state of the art, incorporating diverse insights and relevant results.

Figure 1 illustrates a flowchart identifying the main tasks performed in the development of this state-of-the-art review.

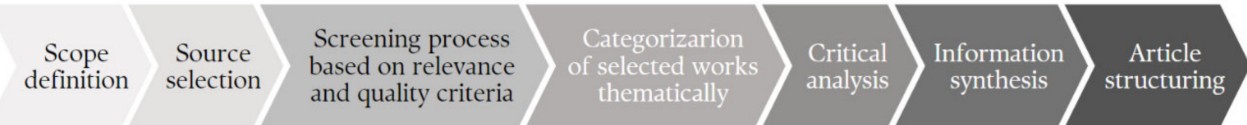

**Figure 1.** Flowchart describing the main steps of the research.

This methodology facilitated the compilation of a state-of-the-art article offering a comprehensive and updated perspective on modular timber construction. It presented key contributions from the existing literature and indicated directions for future research.

The examination of the existing literature resulted in 124 full-text records. Figure 2a displays the distribution patterns of the chosen records after undergoing filtering and the removal of duplicates. Figure 2b showcases the temporal distribution of the selected records.

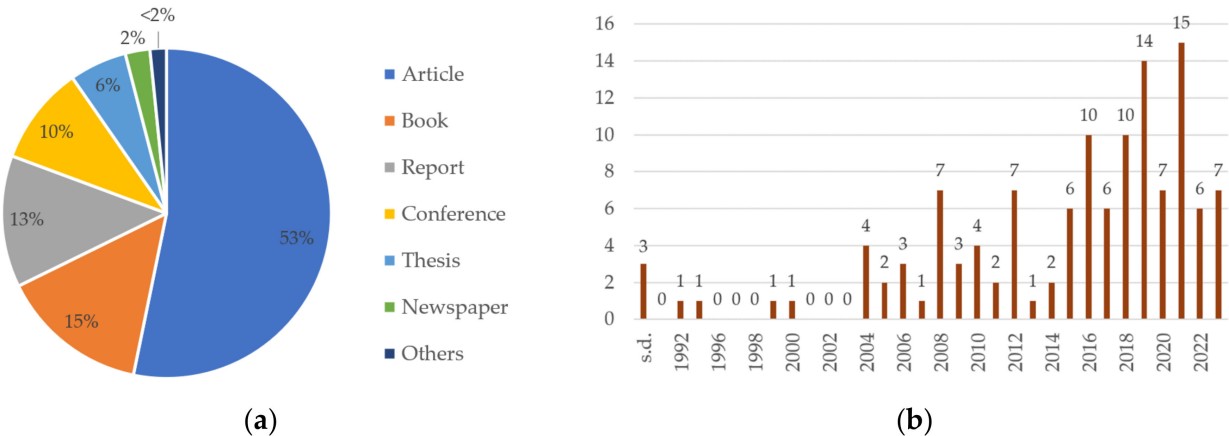

(**a**)   (**b**)

**Figure 2.** Distribution patterns of the selected records of the review (**a**) by record typology; (**b**) by year.

### 3. Wood as a Modern Construction Material for Modular Buildings

The significant role of wood in tall buildings has captured the attention of researchers, construction professionals, and governments worldwide in the past decade. The material's sustainable profile, coupled with its structural properties, such as favorable bending behavior and weight-resistance ratio, has boosted the growing interest in using wood for multi-story structures in the current context.

Addressing the question "Why should one believe it is possible to construct modern tall buildings from wood?", Smith and Frangi [21] conducted a study focused on the specifics of tall timber buildings to identify the key requirements and structural engineering challenges. Analyzing buildings between 10 and 20 stories, they concluded that wood and wood-based products meet the required structural strength functions similarly to concrete and steel. Moreover, wood has the potential to surpass the conservatism of building codes regarding fire safety, ensuring an adequate amount of time for evacuation and firefighting. They emphasized the need for prefabricated systems to ensure that the proper construction details for enhanced mechanical responses and expected functionality (occupant comfort and structural maintenance) can be met.

In recent years, the typology of multi-story timber buildings has gained momentum in European countries. The completion of pioneering projects, such as the Limnologen residential tower in Växjö (2008) and the Stadthaus in London (2009), the world's first nine-story all-timber residential building, indicates a growing confidence in this typology.

Historical large-scale fires played a crucial role in hindering the use of structural wood elements in tall buildings, evident in numerous regulatory barriers present in building codes worldwide. In the European context, despite regulatory requirements not distinguishing between materials, limitations on combustible materials make timber less competitive [22], confining most timber buildings in European countries to a restricted number of stories.

In addition to the standardization of national regulations, government incentives have also made a significant contribution to the consolidation of tall timber construction, especially in leading European countries. Whether to encourage the reduction of greenhouse gas emissions or to promote the timber industry, national economies, employment, or exports from countries with forest reserves, governments have encouraged the use of timber in buildings through environmental policies that promote wood as a building material over steel and concrete. In France, for example, a law was passed requiring the use of timber for

new buildings, except in cases where evidence of incompatibility, usually related to safety, health, or building performance, is provided [23]. In British Columbia, Canada, the Wood First Act [24] aims to foster a wood culture by requiring it to be the first material considered in the early stages of all new provincially funded building projects. In Japan, the Wood First law highlights a preference for wood in the development of public buildings [25]. Programs like Nationella träbyggnadsstrategin and Trästad in Sweden aimed to promote tall wood building construction and were responsible for the rapid increase in interest in the typology in the country [26]. In certain locations, strategic targets were set for vertical wood building construction to reach 50% by 2020 [27]. In Finland, the Ministry of Employment and the Economy led the National Wood Construction Programme from 2011 to 2015, which not only identified projects for large-scale wood construction but also invested in knowledge about high-quality and energy-efficient wood construction, the implementation of new competitive operation models for the construction stage of wood buildings, and the improvement of wood construction training, enhancing efficiency in design, research, and development activities [28].

Currently, educational and professional training mechanisms related to modern wood applications in construction are not at a level comparable to those for structural steel and concrete materials. However, significant initiatives are underway to enhance educational resources and professional training programs, aiming to bridge this gap and foster a more comprehensive understanding of the potential and intricacies of contemporary timber construction practices.

Therefore, the widespread adoption of multi-story wood construction is a result of collaboration between construction and forestry industries, research institutes, and governments. Positive marketing of wood use produced by incentive programs and collaborative research demonstration projects, along with increased customer interest in modern sustainable buildings, has also played a crucial role in this scenario [29,30].

This successful approach is evident in examples such as the e3 building in Berlin (2008), the Stadthaus in London (2009), the Forté apartment building in Melbourne (2012), Treet in Bergen (2015), Brock Commons Tallwood House in Vancouver (2017), and the HoHo hybrid residential-commercial development in Vienna (2020).

It is evident that countries where the use of wood is traditionally widespread, such as Sweden, Canada, the United States, and Japan, adopt new technologies for tall building construction more rapidly due to their experience, confidence, and the availability of the material, along with the established organization of this market. In Sweden, a country with 69% of its territory covered by forests and approximately 90% of single-family homes built using wood as early as 1980, and where timber constructions above two stories were not allowed until 1994, the market share of wood in multi-story buildings increased from 1% in 2000 to 15% in 2012 [26]. In Finland, this increase went from 1% in 2010 to 10% by 2015 [27,31]. In Canada, after the approval for the construction of five to six-story wood buildings in British Columbia, more than 250 buildings were completed in the province [32].

Locations such as the United Kingdom, Germany, and the Netherlands, while needing to address concerns about fire safety and material durability, have shown growth in the widespread use of wood, even for low-rise buildings. In the United Kingdom, the TF2000 project opened new markets for timber-frame constructions up to seven stories high [33] and increased the market share of timber frame from 8% in 1998 to 25% in 2008 [34,35]. In Germany, in 2014, 16.2% of new buildings were built from wood [36]. The use of wood for multi-story buildings was permitted in the country in 2002, and since then, has represented approximately 2% of the multi-story building market [26].

In recent years, there has been a noticeable increase in studies focused on designing multi-story buildings. Most of the existing scientific literature on multi-story timber buildings addresses technical, acoustic [37–39], structural [40–44], energy-related [45], and sustainability aspects [46,47]. However, very few comprehensive and comparative studies have been conducted regarding the practical implementation of wood. Ilgin et al. [48] agree and state that there is a very limited number of studies focused on global trends and

typologies regarding the architectural and structural design considerations of modular timber buildings.

## 4. Engineered Wood Products and Prefabrication

Erecting timber buildings is currently feasible and desirable due to the advantages that wood construction offers, especially when combined with a prefabrication processes.

As a building material, wood presents some deficiencies, such as a high degree of variability and anisotropy, both concerning strength and movements caused by moisture variations, dimensional instability with changes in humidity, and a limited availability of widths and lengths [49]. To overcome these deficiencies and fully leverage the capabilities of new manufacturing methods for a more effective use of the unique characteristics of wood, various products derived from the material were developed during the 20th century, either in the form of panels or beams, known as Engineered Wood Products (EWP).

According to Forestry Innovation Investment [50], EWP refers to value-added wood products made by bonding wood veneers, laminations, strands, or fibers, usually with an adhesive. This manufacturing process generates high-performance and dimensionally stable products for construction projects of different scales. It has benefited from developing and improving adhesive technologies, mechanical connections, and classification and manufacturing technologies [51].

Indeed, the development of EWP has expanded the application of wood in the construction sector. Produced through controlled industrial processes, these products exhibit a high level of quality control and facilitate and endorse the prefabrication process of timber buildings.

Prefabrication involves producing construction units in a controlled factory environment, followed by transporting and assembling these units on-site for the complete construction of the building [52]. The low weight of timber facilitates these steps compared to other structural materials [18]. Within this modern production philosophy, a holistic strategy is necessary, including 3D modelling of the building, well-defined technical systems for resolving complexities in the factory, the use of information communication technology, high levels of planning and process control, and stronger relationships between stakeholders [53]. Prefabrication also impacts the design process, which, in addition to discussions about material performance and structural optimization, must consider solutions regarding construction process efficiency and on-site safety management [18].

Concerning economic sustainability, the construction sector has shown widespread inefficiency and unpredictability in terms of both products and processes due to traditionally employed construction methods [54]. This results from the high degree of complexity in activities and phases, and the competencies of the actors involved, leading to frequent cost increases and delays. On average, the initially estimated time for a project is extended by 20%, resulting in cost overruns of up to 80% [55]. Regarding social sustainability, the sector has the highest rates of workplace accidents and the highest associated costs [56].

In this context, prefabrication has proven to have various benefits for the construction process. The technique, taking place in a factory environment and allowing strict quality controls over prefabricated components, promotes process predictability and enhances the quality and precision of products. A well-managed factory process and well-coordinated efforts on-site also result in improvements in schedules and productivity [57,58] and, consequently, in the reduction of overall project timelines and costs [59,60]. Lehmann [61] estimated a timeframe of three to four months for wood buildings up to nine stories high. According to Richard [62], there is evidence that construction costs can be reduced by 85% when the prefabrication plant is operating at full capacity. This is an economic benefit for builders, who bear lower on-site costs, and for investors, who benefit from the quick marketing of the property and timely delivery [63]. The reduction in on-site activities also contributes to minimizing interference from external conditions and the exposure of components to the elements, preserving their integrity and durability; on-site labor; construction traffic; noise impact and disturbances to the local community; accidents, promoting a controlled and safe construction site; the need for traditional construction

equipment; wet processes; waste, as local factors like overordering, poor execution, and improper material storage are eliminated; and joints, gaps, and penetrations, supporting the creation of a comfortable indoor environment; among others [17,64–68].

The prefabrication construction method inherits concepts that integrate environmental aspects into the product development process which can potentially reduce the negative environmental impacts caused throughout its life cycle phases. Through modular design and dry production (Lean Construction), a wide variety of sustainability goals can be achieved [69]. By promoting the upgrade, adaptation, and modification of components to give them an extended lifespan and influencing component disassembly, prefabrication reduces the industry's environmental burden. Okodi-Iyah [70] proposed that prefabricated construction could save about 44% of the energy used in the construction phase and improve performance by more than 7% during the use phase.

The emergence of fully prefabricated systems has led to production lines consisting of numerous sequential processes, where each station is responsible for a specific function that will contribute to the overall quality and productivity of the assembly line. Traditionally, these processes are completed by conventional means; however, in recent years, an increasing number of these processes are being automated to process materials efficiently and accurately [71].

Automated building processes facilitating efficient in situ resource utilization hold the potential to streamline construction operations in remote areas while offering a carbon-reducing alternative to conventional building practices [72] by minimizing transport distances, costs, and associated pollutant emissions. Importantly, it ensures that the ecological value associated with timber usage remains intact, as it mitigates concerns arising from long-distance transportation. The proximity between the factory and the assembly site enhances operational efficiency and streamlines quality control processes by enabling direct supervision and communication throughout the production cycle. Moreover, this strategy fosters prompt responses to project changes, offering greater flexibility and agility in tailoring components to meet specific project requirements. Unforeseen alterations to the project can be accommodated without significant delays in component delivery, while reduced waiting times for prefabricated components translate to increased on-site productivity. Furthermore, this solution stimulates local economies through job creation or the support of local sawmills and can benefit from the use of native materials. This approach can be partially implemented by focusing solely on essential services completed at the on-site facility. For instance, assembling prefabricated panels into modules on-site helps circumvent logistical challenges and economic concerns associated with transporting fully assembled 3D modules.

There is significant market potential for using wood in various types of buildings using the combination of digital design and CNC (Computer Numeric Control) processing. Digital design and production using CAE (Computer-Aided Engineering), CAD (Computer-Aided Design), and CAM (Computer-Aided Manufacturing) have allowed wood construction to advance to new proportions. During the modeling phase, meticulous attention is given to defining the provisions and dimensions of each element and its components, ensuring compliance with structural requirements while respecting dimensional limitations identified in the design, production, transportation, and on-site application. Employing material optimization strategies is imperative to minimize cuts, thereby reducing production times and waste. This preparatory stage is pivotal for subsequent phases, requiring thorough planning to effectively manage technical and process performance. Following this, the CAD/CAM platform automates the generation of detailed instructions for CNC equipment, facilitating the precise cutting and drilling of elements. Further phases, including lifting, handling, positioning, assembling, and connecting the diverse components, are seamlessly executed by automated systems. This holistic integration optimizes construction processes, bolstering efficiency, consistency, accuracy, and safety, consequently streamlining workflows, and enhancing project outcomes. Innovative connections, modern

wood-based materials, and advanced CNC techniques provide new possibilities and sculpt wood into striking forms [63].

In this context, the Building Information Model (BIM) methodology plays a prominent role, as it not only provides a digital model with parametric information about the building but also promotes the integration and coordination of different materials contained in the project and provides an efficient communication platform for sharing information among various stakeholders [73]. Moreover, Sacks et al. [74] studied the correlations between BIM functionalities and lean construction principles and found strong positive interactions. At present, BIM-enabled tools have widely matured for steel and concrete structures, but they are still not satisfactory for modeling wood structures [75].

Prefabrication represents a unique opportunity for wood technologies to lead the construction sector away from high-consumption, high-waste, and labor-intensive approaches towards a more industrialized and sustainable approach [18,76]. However, despite constant advances in terms of materials, prefabrication technologies, and multi-story timber building assembly methods, the design, as well as the installation procedures for construction services in multi-story timber buildings, lag significantly behind technological advancements observed for other construction materials [77].

## 5. Principles of Modularity in Timber Buildings

The use of modularization as a technique in construction and design aims to achieve design objectives with minimal variation in the building components. This approach is driven by the desire to reduce the number of distinct building elements in a project, promoting standardization and efficiency.

While in freeform construction buildings are designed with little or no adherence to regular geometrical shapes or standardized components, allowing for highly customized and innovative architectural designs but demanding more time and resources, by adopting the recent adaptative modular construction technique, designers and builders seek to balance standardization and adaptability, aiming for an efficient and versatile system. This approach aligns with broader trends in modular construction and prefabrication, emphasizing the benefits of standardized, repeatable elements in the built environment, such as quality, speed of construction, and cost-effectiveness.

The discrete design agenda has played a pivotal role in transitioning from strict modularity to adaptive modular construction. In this approach, fully functional and intricate buildings are assembled from serially repeating, rearrangeable sets of generic discrete elements, each treated as an individual entity with clear boundaries [78]. This approach diverges from the serialized production of identical units or generic solutions; instead, it relies on the combination and variation of purposefully designed parts to achieve customization and adaptability through scalable principles [79]. In this particular context, the kit-of-parts methodology emerges as an alternative strategy within sustainable design. This methodology revolves around the concept of a pre-designed collection of discrete building components that can be assembled in various ways into a finished building [80]. The kit's components are systematically generated using a combinatorial approach, affording a broad spectrum of design possibilities [81].

This design and construction methodology entails deconstructing a building or project into modular components, each conceptualized as integral elements of a standardized kit. These components are pre-designed, prefabricated, and mass-produced, facilitating seamless assembly, offering flexibility, and fostering efficiency in the construction process.

However, a notable challenge in expanding the application of the kit-of-parts approach lies in the absence of universally applicable toolkits and libraries of configuration tools. Consequently, manufacturers face the imperative to escalate their efforts to craft solutions that seamlessly integrate into the supply chain from design to production [82]. In the kit-of-parts approach, the overarching objective is to establish a system wherein these components can be amalgamated and reconfigured to cater to diverse needs, thereby offering a solution characterized by versatility and adaptability [81].

This approach has been employed in various architectural and construction projects, ranging from residential buildings to commercial structures. It aligns with modular construction and prefabrication principles, emphasizing standardization, efficiency, flexibility, and sustainability in the built environment.

## 6. Taxonomy of Modular Timber Products

Modular timber products' taxonomy involves categorizing them based on their characteristics, functionalities, and applications.

The fundamental load-bearing element in modular constructions is the modular unit. Different structural configurations of modules designed for modular high-rise buildings have been developed and can be broadly classified [83]. However, the literature does not consistently agree upon the structural categorization of mass timber buildings. As Salvadori [84] elaborates, Mass Timber Buildings (MTBs) can take shape through one-dimensional (1D) or two-dimensional (2D) structural elements (vertical and horizontal) or by employing three-dimensional (3D) modules, where the walls and floors are pre-assembled. This author established 32 categories that intricately combine and cross-reference the primary types of structures and materials used in various systems. In contrast, Svatoš-Ražnjevi'c et al. [85] introduced a classification system based on the predominant usage of 1D, 2D, or 3D timber elements in construction and their respective combinations. The established categories include the following (Figure 3):

- 1D frame structure;
- 2D bearing wall;
- 3D volumetric modules;
- Combination or hybrid.

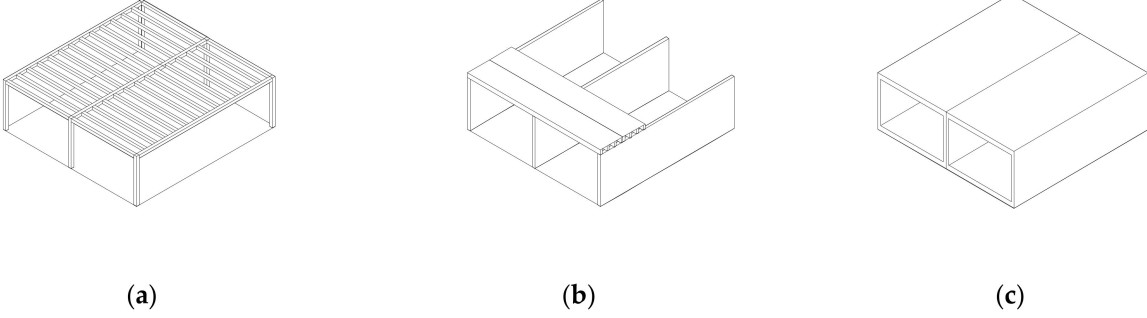

(**a**)  (**b**)  (**c**)

**Figure 3.** Modularity in (**a**) 1D elements; (**b**) 2D panels; and (**c**) 3D volumetric modules. Adapted from Kaufmann et al. [86].

According to Svatoš-Ražnjevi'c et al. [85] and Kuzmanovska et al. [87], panel and space module systems were the most common systems from 2000 to 2010, while from 2009 to 2011, there was a significant increase in frame structures.

The research findings of Svatoš-Ražnjevi'c et al. [85] establish that panel and volumetric module systems are predominantly employed in mid-rise projects, typically reaching up to 10 stories in height. Although taller structures exist within panel and volumetric systems, combination systems emerge as the preferred choice for projects exceeding 15 stories, including notable specimens ranging from 29 to 80 stories in height [85].

This trend aligns with the observation that a significant majority of the proposed but unbuilt projects considered by Svatoš-Ražnjevi'c et al. [85] opt for frame structures. These unbuilt projects encompass some of the tallest Modular and Sustainable Tall Buildings (MSTBs). However, it is crucial to acknowledge that this prevalence of frame structures in unbuilt projects may be influenced by the data sources and the limited visibility of modular projects in mainstream literature and timber databases.

### 6.1. One-Dimensional/1D Frame Structure/Linear Systems

One-dimensional (1D) frame structures, also known as linear systems, play a crucial role in timber buildings, offering a versatile and effective approach to structural design.

They are relatively simple in their assembly, making them practical for various building designs. Moreover, these structures offer versatility in terms of configuration, allowing architects and engineers to adapt them to different architectural styles and functional requirements.

One-dimensional elements form beam–column structures, slab–column structures, and exoskeleton structures, where the vertical supports are often limited to the exterior and fixed to a core, contributing to structural stability and architectural aesthetics. According to Svatoš-Ražnjevi'c et al. [85], one-dimensional element structures are primarily used for five to eight-story projects. Nearly one-third of the projects comprise buildings with more than nine floors (7% with more than twenty floors).

Although structures with one-dimensional elements may consist of just an articulated assembly of linear timber pieces, additional bracing systems, such as shear walls, diagonal EWPs, transverse metal bracing, and metal beams, can be added to the structures to ensure their lateral stability.

For these structures, various combinations of EWPs can be employed, such CLT slabs, ribbed slabs, or composite floors (CLT–concrete or glulam–concrete), to create floor slabs [85].

Using traditional column and beam structural forms in 1D frame structures may be considered effective and aligned with the structural design criteria in mass timber projects. This approach optimizes the use of timber elements and meets the sustainability goals associated with modern construction practices. On the other hand, in modern timber modules, adopting the conventional column and beam structural form may be considered excessive and is seldom utilized [81].

### 6.2. Two-Dimensional/2D Bearing Wall/Panelized Systems

A 2D panelized solution resembles the flat-pack assembly approach commonly employed in home furniture [88]. These systems involve the use of prefabricated panels that serve as both vertical load-bearing walls and horizontal floors or roofs, manufactured offsite and then transported to the construction site for assembly. This approach offers efficiency, speed, and versatility in construction.

These panels can be categorized into various types based on their specific functions within the building structure. They typically consist of a framework with integrated sheathing and insulation. They are also often made from EWPs, in particular CLT or other prefabricated timber panels. Where necessary, panels include conduits for essential services such as heating, ventilation, air conditioning (HVAC), and plumbing. These conduits can be seamlessly linked using standard connectors [88]. Two-dimensional panel systems primarily consist of medium-height projects up to ten floors. Medium-height projects ranging from five to eight floors are the majority, followed by low-rise building projects of three to four floors [85].

A distinctive feature of the modular product lies in its flexible design. For example, components can be combined to accommodate windows and doors in various locations and sizes [81]. Two-dimensional panelized solutions provide greater flexibility than three-dimensional modules [88]. For instance, large open-plan offices may be unsuitable for single 3D modular elements. The relevance of 2D panels extends to high-end residential projects, including single-family homes and apartments. In such projects, where differentiation is crucial and the ratio of wet to dry areas is lower, 2D panels offer a more adaptable and suitable solution [88].

The onsite assembly process for flat-pack panels is simpler than traditional construction but more intricate than assembling 3D modules. It does require additional onsite assembly and internal finishing. However, transporting the panels is much more convenient than larger 3D modules. Using flat-pack panels allows materials covering a significantly larger floor area to be transported at once. As stated in [88], it costs approximately 8 USD

per square meter of floor space to transport 2D panels around 250 km, whereas the cost is nearly 45 USD per square meter for the 3D equivalent.

In contemporary timber construction, 2D bearing wall systems have gained popularity due to their efficiency and sustainability. The prefabrication of panels allows for precise engineering and quality control, resulting in structures that meet modern performance and energy efficiency standards. Additionally, these systems contribute to the growing trend of modular and offsite construction practices in the timber building industry.

### 6.3. Three-Dimensional/3D Volumetric Module/Modular Systems

Modules are autonomous three-dimensional units or partially assembled sections. Their versatility allows for repetition through stacking or joining them side by side, facilitating the extension of spaces [89]. This approach is most suitable for projects characterized by a high level of repeatability and a significant ratio of wet to dry rooms. It is important to note that repeatability in this context does not imply uniformity in appearance. Instead, a range of standardized modules can be assembled in various configurations to achieve a customized and diverse result [88]. According to Svatoš-Ražnjevi'c et al. [85], only 6% to 10% of the volumetric panel and module projects exceed nine floors, with a maximum height of 19 and 15 floors, respectively.

In contrast to panelized or linear types of prefabrication, modular construction involves the integration of most interior and exterior finishes within the factory setting [83,90]. In fact, the module represents the most complete form of prefabrication, with approximately 95% of the essential kitchen and bathroom facilities, storage amenities, and living spaces being fully fitted in the factory and subsequently sealed for transportation. This approach is particularly advantageous for rooms requiring intricate finishing, especially wet rooms such as bathrooms and kitchens [88]. The primary advantage of modular building is that the structure becomes immediately ready for use once power and water facilities are connected [89].

In addition to maximizing factory production, modular construction enjoys the advantages of being unaffected by weather conditions, achieving precision through advanced tooling, maintaining high-quality control, streamlining assembly lines with semi-skilled labor or automation/robotics, flexible grouping geometries, and benefiting from bulk purchasing of components [91].

Nevertheless, it requires significant (though not prohibitive) transportation costs, mainly because costs are primarily associated with volume and secondarily with weight. This implies that transporting empty modules (such as living areas, bedrooms, etc.) incurs costs comparable to fully equipped modules with value-added content (kitchen, bathroom, services, etc.) and is subject to vertical transportation restrictions [91]. This either raises the transportation costs for larger units or constrains the dimensions of modules, making 3D volumetric solutions particularly well-suited for applications such as hotels, hostels, or affordable housing [88].

Modules are designed for ease of assembly, yet compared to other prefabrication types, this approach is deemed the most cumbersome and technically challenging [89]. Onsite assembly in modular construction entails lifting the modules into place using cranes, bolting them into position and connecting services such as electricity [88]. During the lifting process, the modules are subjected to structural forces, particularly vertical bending and shear, which differ from the forces they experience after installation. The necessity to reinforce the modules for lifting and the doubling of floor, roof, and wall frames at the connection points leads to increased quantities of structural material [89].

In a case study, construction using the volumetric modular technique was completed 16% faster than that using the 2D approach. Despite the apparent transportation difficulties in modular construction compared to the 2D method, the volumetric construction method saves more time without sacrificing quality [92]. When evaluating the difference between the three options (1D, 2D, and 3D) for an affordable housing unit with four floors, Bertram [88] found that a 2D solution could be 17% cheaper than a traditional approach,

while a 2D and 3D hybrid solution lowers costs by 20%, and a 3D solution by 24%. This would vary by project, but these estimates indicate the scale of potential savings.

Three-dimensional (3D) volumetric module systems in timber buildings represent an advanced approach to construction, leveraging prefabricated modules encompassing complete building volumes. These systems provide a holistic and efficient solution for assembling entire structures with integrated components.

### 6.4. Hybrid Systems

Hybrid systems in timber buildings refer to the integration of different structural approaches within a single construction project, combining elements from one-dimensional (1D), two-dimensional (2D), and three-dimensional (3D) systems. This hybridization allows for a tailored and optimized solution, leveraging the strengths of each system to address specific structural and design requirements.

In hybrid structural systems, categories are combined in multiple ways. Projects, therefore, fall into one of the following categories [85]:

- The upper and lower sections of the building modules are constructed with different structural characteristics;
- Different zones of the building floor plan are constructed using distinct systems.

In modular construction, it is feasible to employ a combination of 3D modules and 2D panels within a project or even integrate these approaches with traditional onsite construction, especially for components like basements and the first floor of larger projects [83,88,90]. An example is the construction of areas resembling corridors, often seen in hotels, that can be achieved by repetitively utilizing 3D units. In contrast, the corridors can be produced as planar elements or as extensions to room modules [93].

Typically, wet areas, such as bathrooms, are manufactured as bathroom pods, while the rest of the building is constructed using 2D panels. This strategic approach optimizes the construction process for different building areas, enhancing productivity for bathroom spaces and providing maximum flexibility for other areas. However, it is important to note that the manufacturing process required to deliver both solutions becomes more intricate, as does the coordination of the overall supply chain [88].

According to Svatoš-Ražnjevi'c et al. [85], hybrid structural systems are primarily composed of combinations of 1D elements and 2D elements, followed by the combination of 3D modules and exoskeletons, timber frames and solid wood slabs, 2D panels and 3D modules, and finally, projects combined with additional external structures such as balconies. Based on Žegarac Leskovar and Premrov [94], aiming to increase the height of buildings, hybrid systems of beams and columns associated with solid panels (CLT slabs or walls) are commonly used.

In contemporary timber building projects, adopting hybrid systems reflects a holistic and pragmatic approach to construction. This integration of 1D, 2D, and 3D elements allows for innovative and efficient solutions that meet the demands of modern design, structural integrity, and sustainability goals.

## 7. Manufacturing, Transportation, and Installation Dimensions

Dimensional constraints play a crucial role in the manufacturing, transportation, and installation of prefabricated wood panels and modules. In the manufacturing process, limited dimensions may be determined by the factory's capacity and the equipment used. Transportation constraints are related to the dimensions allowed on public roads and transport regulations. Additionally, the choice of transportation modes, such as trucks or containers, may be influenced by the dimensions of prefabricated components. During installation, dimension constraints impact the logistics of the construction site, including access to the site, available space, and the ability to handle bulky components. Therefore, careful consideration of these constraints is essential to optimize the prefabricated wood construction process.

The dimensions of 2D panels are primarily constrained by production and transportation considerations, given that their geometry and lightweight nature do not pose significant challenges during the lifting and installation phases. Typically, the cutting machinery utilized allows for dimensions of $3.50 \times 9.00$ m$^2$. When considering a standard truck, the panels are confined to the following specifications:

- Length: 13.50 m;
- Width: 2.45–2.48 m;
- Height: 2.50–3.00 m;
- Load weight: 24.00–25.50 tons.

Transportation is a critical and design-defining phase in the construction of room modules. Special forms of transport, subject to approval, are typically utilized for this purpose. However, the maximum size of room modules is predominantly determined not only by the production facility and on-site conditions but also by the transport capacities offered by the truck.

Standard semi-trailer trucks, with a loading length of 13.50 m, are suitable for most projects, but longer vehicles can handle considerably larger lengths with manageable additional costs, reaching a critical point at around 15 m. For smaller dimensions, up to approximately 6.50 m in length, transporting two modules in one vehicle is feasible. Most roads recommend a 4.30 m maximum total transport height. With standard loading platform heights ranging from 0.90 to 1.10 m, a minimum height of 3.20 m is retained for the modules. Special low-bed semi-trailer trucks enable the transportation of room modules with a maximum height of 4.20 m, subject to specific limitations regarding the length of the cubicles. Maintaining a total width of 3.25 m mitigates transportation issues, facilitating navigation through unforeseen construction sites. For transport widths exceeding 4.00 m, commissioning a transport study to analyze the route is advisable. The average costs for a transport route spanning several hundred kilometers are estimated to be around 5% of the pure construction costs, with the complexity of transport (police escort, etc.) contributing to price fluctuations of up to 30% [95].

Timber room modules, typically weighing 350–400 kg/m$^2$ in standard sizes, may approach the capacity limit of common semi-trailer trucks when exceeding 20 tons. Room modules are generally lifted using mobile cranes. If the modules are sufficiently rigid, they can be directly mounted on the crane hook using steel cables at four points. For less rigid modules, such as those made of timber-frame elements, an additional lifting structure, often in the form of a robust steel frame, may be necessary to provide additional attachment points. This particularly applies to large modules or those with a lower rigidity than that of cross-laminated timber [95]. When choosing transportation via maritime containers, it is crucial to consider specific dimensions: a length ranging from 5.90 m to 12.03 m, a width of 2.35 m, and a height of 2.39 m. Additionally, it is important to consider door opening dimensions when applicable.

Li et al. [81] assessed design details for the dimensions, weight, and floor area of volumetric timber items by analyzing modular products made by eight suppliers and the modular products used in 60 modular timber projects.

The length of a typical modular unit is often from 6 to 12 m [96]. The significant variation in lengths is a result of the adaptable design. Shorter lengths are derived from functional modules, such as bathrooms, separate from residential units. Modules less than 5 m long are functional or flexibly designed modules. Modules longer than 10 m are generally made for apartments. The most variable building types in module length are apartments and social housing structures, followed by hotels. This can be attributed mainly to the combination of modules tailored to fulfill the diverse demands for different types of living spaces. Shorter lengths are typically fused to generate more expansive areas. Shorter lengths are typically merged to create larger areas. For student housing, nursing homes, and hospitals, the modules do not typically use a combination design method. Nevertheless, the dimensions of the module may still vary depending on the application;

for instance, certain modules may need to be configured without obstacles and with slightly increased lengths to facilitate wheelchair access [81].

Most timber volumetric modules have a width ranging from 2.50 to 4.00 m. The module's intended function has little significant influence on its width. However, the modules utilized in apartments and social housing encompass an extensive range. Hotel modules exhibited the highest degree of variability in terms of width. Transportation conditions primarily contribute to the concentration of width values between 2 and 5 m. Typically, modular rooms exceeding a width of 5 m are constructed by integrating multiple 3D modules, commonly found in offices and educational facilities requiring more space [81].

Ceiling height is also a significant architectural consideration in timber buildings. More than half of the modules that were investigated by Li et al. [81] had heights in the interval [3.00, 3.50) m. The MEP (mechanical, electrical, and plumbing) system area is positioned either above the ceiling or below the floor of the room module, depending on the construction technique. Ali and Armstrong [97] proposed a height range of 2.40 to 2.70 m from floor to ceiling (including finishes) for residential purposes, irrespective of the country or region. The Tall Buildings Statement of London [98] recommends a ceiling height of 3.20 m, reflecting the need to accommodate facilities and services. According to Ilgin et al. [48], measurements considering the distance between the floor of one level and the floor of the next range from 2.81 to 3.30 m, with an average of 3.00 m. Only two out of the thirteen buildings they analyzed exceeded the 3.20 m suggested by the Tall Buildings Statement of London [98].

The largest floor area of 50 $m^2$ is used for hotels and residential buildings. The standard floor space for volumetric modules ranges from 15 to 25 $m^2$ [81].

Timber modules can be divided into functional and room modules based on the difference in mass. Functional modules encompass various facilities, such as lavatories (including saunas) and equipment rooms. The mass of timber room modules falls into the range of 5 to 15 t, and the mass of an equipment room is less than 25 t [81].

The dimensions of a standard modular unit are constrained by transportation and lifting requirements. In addition, the maximum lifting capacity of most tower cranes used in building projects is around 20 tons, and the cost of a tower crane with a lifting limit of over 20 tons might increase by up to 60%. Hence, transportation and lifting requirements serve as the primary limitations in establishing both the module size and hoist weight. In the case of high-rise buildings, the modules at the base experience substantial vertical loads. This leads to an increase in their member sizes and hoisting weight, thereby reducing the modules' life span. However, using short modules will result in an increased number of inter-module joints, so the benefits of modular construction are not fully maximized [83].

## 8. Layout Design

The arrangement of a building is influenced by its purpose, structural design, seismic considerations, topographical constraints, and the specifications of various design alternatives [81].

Although most of the literature focuses on technical and structural aspects, the few architectural design aspects examined have been limited to the geometric description of buildings, lacking a comprehensive exploration of the variety of forms and spatial organizations in timber construction. According to Žegarac Leskovar and Premrov [94], early modern timber buildings had a relatively regular geometry with symmetrical and repetitive floor plans. However, contemporary buildings exhibit more geometric variations, including vertical irregularities due to variations in floor elements, distortion of the basic structural grid, and asymmetric configurations of facade openings such as windows and balconies [87]. These architectural expressions pose technical challenges and require customized solutions.

Contrary to current trends in concrete and steel buildings, timber structures show a prevalence of rectilinear plans and regular extrusions with a low degree of innovation, relying on the simple use of timber rather than its expression as the primary material—a

marketing strategy. Exceptions are rare, and more complex projects often involve hybrid structures or small footprints adapted to specific geometric constraints in medium- to high-density urban locations.

Svatoš-Ražnjevi'c et al. [85] reported a continuous increase in building height. The first five-story building was introduced in 2004, followed by the first six-story building in 2006, seven and eight-story buildings in 2008, a nine-story building in 2009, a ten-story building in 2012, and so forth.

The analysis of floor plans and building silhouettes reveals that most projects are rectangular, with the only recurring alternative being the "L" shape. Projects using two-dimensional panels include examples that deviate the most from orthogonal shapes, followed by projects using one-dimensional elements. Volumetric projects have no examples of non-orthogonal forms. Three-dimensional module projects present the smallest range of typologies, with only rectangular and similar examples. According to Li et al. [81], there are currently 21.77 modules on average per floor.

Furthermore, over 80% of projects are symmetrical, with 3D module projects having the fewest non-symmetrical designs, while combined systems comprise the majority of this typology. As an example of symmetry, Ilgin and Karjalainen [99] highlight a preference for centrally located cores, followed by peripheral cores, mirroring the trends seen in concrete or steel buildings. The core's location, and consequently those of the elevator and stairs, is a crucial architectural parameter for multi-story buildings, following the classification defined by Ilgin et al. [48], as represented in Figure 4.

The benefits of a central core include aspects such as structural support, compactness (allowing openings in the facade), and improved safety performance for fire stairs (which may have contributed to the prevalence of this typology). However, Kuzmanovska et al. [87] conclude that most central circulation schemes are independent of the core's location. On the other hand, the disadvantages of peripheral core configurations include low space efficiency, such as longer corridors and longer evacuation distances in the case of fire. The same applies to external core configurations, for which there are no records of their use in multi-story residential timber buildings.

Furthermore, regular extrusions with a flat roof dominate all structural systems. There is less variation in volume in 3D module projects, which do not have irregular examples. In contrast, 1D and 2D element projects show much more volumetric variations. Even within regular extrusions, variations in the roof are more common.

Through the work of Ilgin and Karjalainen [99], it was concluded that prismatic forms with rectilinear plans and regular extrusions are the most common and occur in eight out of ten case studies. The reasons presented by the authors encompass ease of construction, practicality, and efficient use of interior space (especially in rectangular floor plans). These are also the most used forms in residential buildings made of concrete or steel, indicating that this trend is not dependent on the structural material (Figure 5).

The grid is the dominant structural ordering system, followed by the linear strategy or a combination of both. One-dimensional element projects generally follow grid systems, while two-dimensional panel and three-dimensional module projects are primarily ordered by linear arrays. Combined systems have two dominant strategies: grid and linear array (Figure 6) [85]. According to Hua et al. [100], the symmetry in a grid is essential to the modularity in design and construction. There are multiple choices in terms of the symmetry in an architectural floor plan, but these decisions determine different prefabrication and logistics conditions.

Balconies and facades are other important design elements worth considering in a project. Both elements can be constructed on-site or assembled using modules in an off-site factory. The selection of materials is not restricted to timber and can be accomplished aesthetically to hide the appearance of modular construction [81].

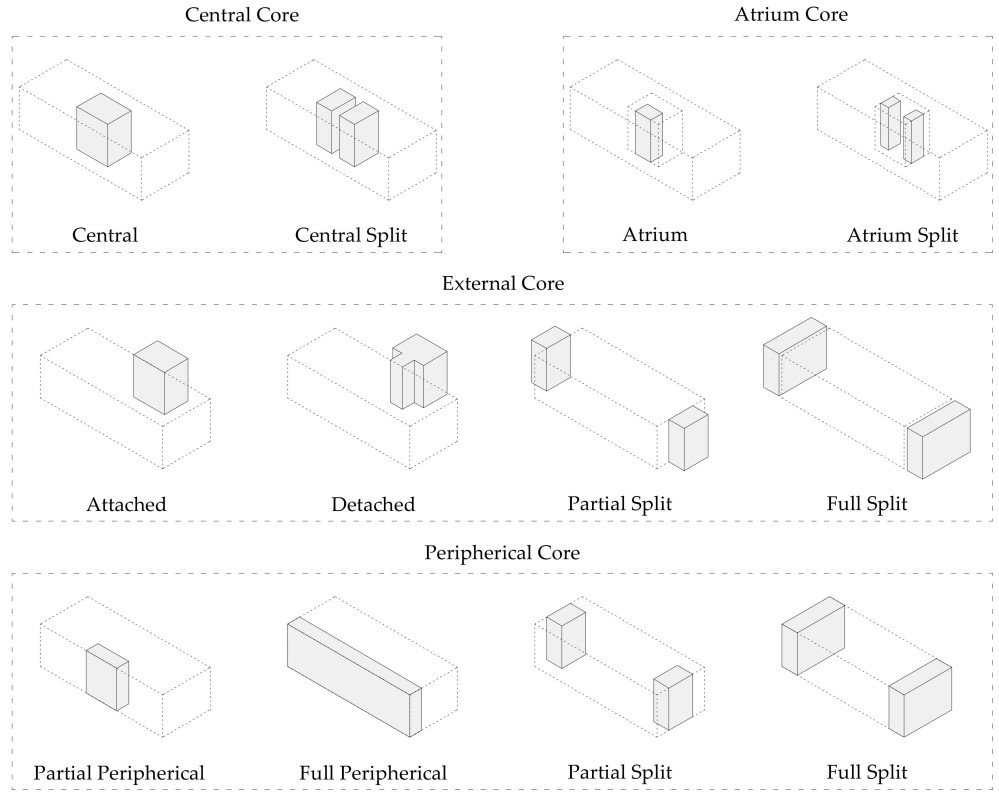

**Figure 4.** Classification of central arrangements for tall buildings. Adapted from Ilgin et al. [48].

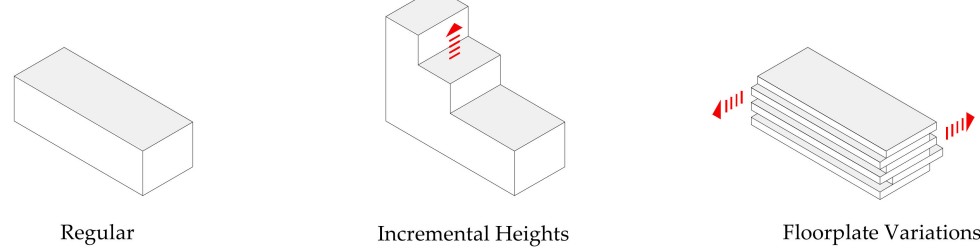

**Figure 5.** Examples of volumetric extrusions. Adapted from Svatoš-ražnjević et al. [85].

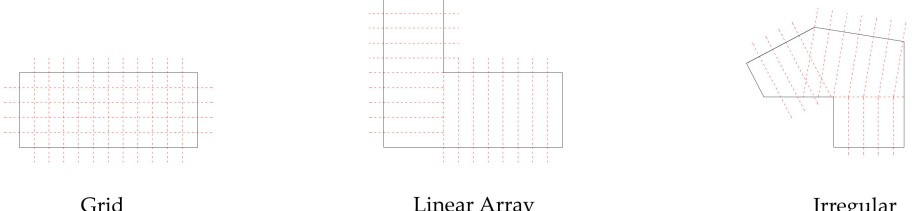

**Figure 6.** Examples of structural ordering systems. Adapted from Svatoš-ražnjević et al. [85].

The materials used for facade cladding encompass a range of options, such as timber battens, stainless steel, fiber cement board, glass panels, copper, photovoltaic panels (BIPV), and aluminum. Timber facades, where the timber elements lack protection, leading to an increased risk of degradation (use class 3, corresponding to service class 3 of Eurocode 5 [101]), should be constructed using hardwood species with inherent durability or treated appropriately. A greater diversity of facade forms can be provided using coated and corrugated aluminum panels. All the aforementioned facade designs are well-suited for on-site installation. Moreover, the building's facade can be selected from the external surfaces of the wood modules or chosen to impart a non-modular appearance to the modularly

constructed buildings. In Li et al. [81], the term "non-modular façade" refers to a situation in which the material completely hides the modular connection details.

## 9. Building Occupancy and Use

The most common program in timber building construction is residential, comprising nearly half of all projects [85,99]. However, Kuzmanovska et al. [87] noted a shift away from entirely residential programs associated with a growing preference toward mixed-use programs. Currently, projects geared towards the development of modular timber systems are noticeable, catering not only to residential buildings but also extending to hotels, senior residences, student accommodations, hospitals, and schools.

Svatoš-Ražnjevi'c et al. [85] concluded that office buildings constitute the majority of one-dimensional systems. On the other hand, residential buildings (whether for social housing, student housing, or multifamily housing) comprise most of the panel systems. In 3D module construction, the even distribution between commercial and residential spaces results from a high number of hotels and hostels being counted as commercial spaces and a lower number of spaces being designated for offices. Most residential program projects are for affordable housing. Other available options include nursing homes, apartments, and various types of accommodations. School projects also have a significant presence in 3D module buildings. Residential programs constitute the majority of hybrid systems.

## 10. Structural Systems for Modular Buildings

### 10.1. Building Level

Structural systems for modular high-rise buildings are pivotal in ensuring stability, safety, and efficiency. Various approaches are employed to address the unique challenges posed by the vertical expansion of modular structures. Here we present the key considerations and structural systems commonly used in high-rise modular buildings.

Svatoš-Ražnjevi'c et al. [85] provided a complete overview of the sub-categories of structural systems that appeared during their project analysis (Table 1).

Ilgin et al. [48] identified six structural systems for tall buildings in their study (Figure 7).

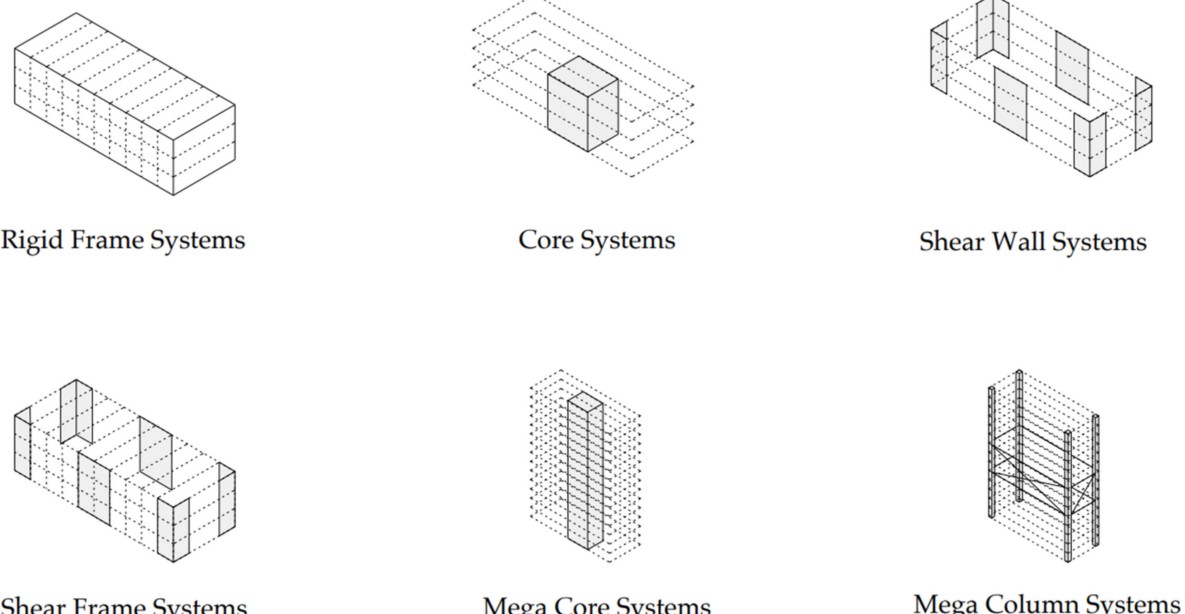

**Figure 7.** Structural systems for tall buildings. Adapted from Ilgin et al. [48].

**Table 1.** Sub-categories of structural systems. Adapted from Svatoš-ražnjević et al. [85].

| Type of Structural System | Sub-System |
|---|---|
| One-dimensional Frame structure Linear systems | Exoskeleton Post-and-beam Post-and-beam with linear bracing Post-and-slab Post-and-slab-band |
| Two-dimensional Bearing wall Panelized systems | Crosswall and party wall Honeycomb Panel + beams Panel + box beams Panel + truss Panel + beams + columns Panel + columns Panel + external frame (balconies) |
| Three-dimensional Volumetric modules Modular systems | Space modules Space modules + external frame (balconies) |
| Combination Hybrid systems | Frame + panel Frame + space modules Exoskeleton + space modules Light frame + mass timber Panel + beams + external frame (balconies) |

Within the domain of stacked Prefabricated Volumetric Modular Systems (PFVMS) buildings, a vertical augmentation of modular units facilitates the establishment of multi-story structures. This modular system optimizes vertical space utilization without encroaching upon ground space [102]. Particularly prevalent in low-to-mid-rise residential apartments, this solution combines modules in a linear or planar fashion, allowing for the pursuit of space efficiency [93]. This approach is applied similarly regardless of the chosen material. The modules in this stack are interconnected using rigid intermodular connections [103] (Figure 8a). Nevertheless, relying solely on intermodular connections may not effectively contribute to load resistance, necessitating the incorporation of stability technologies to withstand lateral forces in high-rise Modular Steel Buildings (MSBs). These stability systems may include steel braces, steel-plate shear walls, multi-column walls, base isolation systems, concrete cores, and moment frames. The development of these stabilization technologies can follow traditional or modern modular methodologies, utilizing precast, steel, composite, or steel–concrete hybrid materials [104].

Another option is a superstructure of beams and columns as the main structure into which prefabricated modules are inserted, avoiding direct overloading on the underlying elements (Figure 8b). This approach is common practice in modular constructions, where the support infrastructure is established using vertical (columns) and horizontal (beams) elements. Prefabricated modules, manufactured off-site, are then positioned, and secured within this framework. This configuration offers advantages as it allows for a more efficient assembly of modules, simplifies the construction process, and provides flexibility in the design of interior spaces. The beams-and-columns superstructure provides the necessary stability and the ability to support vertical loads, ensuring that modules can be integrated safely and effectively. This methodology is commonly employed in modular construction projects, where coordination between the superstructure and prefabricated modules is essential to ensure the structural and functional integrity of the building. However, this solution also presents challenges, including additional costs due to structural redundancy and complexities in the connections between the megastructure and the modules.

The third structural solution involves anchoring the modular units directly to the central core of the building, contributing to structural cohesion (Figure 8c). This approach

enhances stability and ensures a robust connection between the prefabricated modules and the building core, providing a secure and integrated structural system.

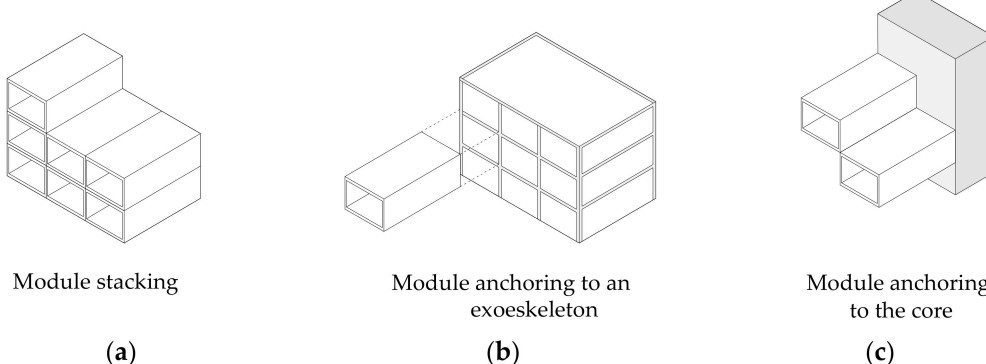

| Module stacking | Module anchoring to an exoeskeleton | Module anchoring to the core |
| :---: | :---: | :---: |
| (**a**) | (**b**) | (**c**) |

**Figure 8.** Structural solutions to support volumetric modules: (**a**) module stacking; (**b**) module anchoring to an exoskeleton; (**c**) module anchoring to the core.

Timber has a strength (parallel to grain) that is similar to that of reinforced concrete; hardwood is slightly stronger, and softwood is slightly weaker. Nevertheless, timber cannot rival the compression strength of modern high-strength concrete. Timber is less stiff than concrete, and both materials are far less stiff or strong than steel. However, timber has a low density compared to these other conventional structural materials. Due to this, when considering strength-to-weight and modulus-to-weight ratios, softwood demonstrates a comparable performance to steel. This implies that timber is notably structurally efficient in constructions, or components of constructions, where a significant portion of the load to be withstood is the self-weight of the structure. Examples are roofs, some bridges, and tall buildings' gravity load-resisting systems [105].

In designing tall timber buildings under specific circumstances, such as in seismically active regions, an additional challenge arises beyond the effects of gravity and wind loading. It is related to the dissipation of seismic energy within the structure, as timber is a non-ductile material. In a full-timber construction, the dissipation of seismic energy occurs at the connecting points between the timber elements, such as nails, screws, bolts, hold-downs, etc. Structural timber members can be combined with other materials, such as steel and concrete, to cope with the disadvantages found within tall buildings' structural and seismic design that have been mentioned [106].

In multi-story timber construction, access cores (such as emergency exits and those used to brace buildings), firewalls, or entire reinforced concrete base stories are now often combined with timber structures, and steel-reinforced concrete frames or compartmentalized structures with a building envelope made of highly insulated timber panel elements are increasingly being used [86].

Combining concrete with timber in multi-story buildings offers several advantages over solid wood structures. This combination allows for longer spans, providing better performance in terms of vibration, deflection, and sagging due to increased structural stiffness. It enhances fire safety and soundproofing and minimizes vertical subsidence. Additionally, the concrete layer protects the underlying timber during construction and potential interior leaks. However, composite timber–concrete structures can be heavier if additional mass is not needed for soundproofing purposes [86].

Steel, known for its structural strength, is often used in situations that require support for long spans and heavy loads. By incorporating timber in this context, design flexibility is expanded, allowing for the creation of innovative, sustainable, and aesthetically pleasing structures. Structural efficiency is then optimized by leveraging the strength of steel against compression and tension forces, resulting in more robust and durable structures. A steel frame is also sometimes combined with a timber secondary structure to transfer heavy loads in multi-story timber buildings [86].

In general, most timber buildings exhibit a hybrid and complex structure, as mentioned by Salvadori [107]. With only 78 examples, fully timber constructions are uncommon in all of the structural categories considered, representing 22.6% of all projects, while the remaining structures are hybrids. Most hybrid structures in timber buildings (54%) combine timber and concrete. Timber–concrete–steel structures account for 13.1%, and timber–steel structures for 5.4%.

The classification also considers the presence of these materials in subsequent structural components [85]:

- A podium or plinth (only the ground floor or a set of floors in the lower part of the building), which can be made of concrete or sometimes steel;
- The core, which can be made of timber (CLT or LVL), concrete, or an articulated set of one-dimensional metallic elements;
- Floor slabs, which can be made from a range of EWP or in a composite solution (timber with a layer of concrete or embedded precast concrete or steel beams);
- Lateral bracing and vertical/horizontal structural elements, which can be made of timber or steel;
- Others, mostly comprising external structural elements such as stairs and circulation areas.

According to Kuzmanovska et al. [87], the use of concrete cores has increased over time (from 38% in 2009–2013 to 57% in 2018–2020) and is more prevalent in structures with more than 16 floors. Conversely, the use of CLT cores has decreased (from 63% to 43% over the same period). Additionally, their study revealed a discrepancy among nine-story structures: an increased use of CLT cores and a decline in concrete cores. However, there was a noticeable growth in the use of steel cores between 2017 and 2018, indicating a transition to hybrid models as systems evolve. The study's results make it challenging to determine whether the increased use of concrete cores results from taller building heights, increased beam–column system construction (which may require a core with more mass than CLT can provide), the trend of hybridization, or a combination of these factors.

Kuzmanovska et al. [87] also noted a growing trend in the use of podiums (25% in 2009–2013 to 60% in 2018–2020). Interestingly, this measure seems not to correlate with a building's height or its program.

According to Ilgin et al. [48], using concrete podium structures offers advantages, such as accommodating ground-level commercial zones, providing open areas and large openings, and creating fire-resistant enclosures for substantial mechanical/electrical services and equipment. The timber structure is also outside the range of water splashing and earth moisture, and level transitions between inside and out can be created while providing structural protection for the timber without complex detailing. This type of structure also enables planners to create emergency staircases leading outside and access routes for emergency services without having to add special separate fire safety measures [86].

On the other hand, the motivation for using concrete in the core may be based on promoting lateral stiffness and strength in the structure, benefiting from concrete's inherent fire resistance, and reducing wind-induced sway, a commonly reported issue in tall buildings. In the case of Brock Commons Tallwood House, it was found that using concrete in the core also simplified the project approval process [48]. Nevertheless, integrating various materials in vertical structural components may also lead to challenges or issues. Concrete staircases, lift shafts, or similar structural components must usually be built at an early stage of construction and drying times and formwork processes can significantly extend construction times. The dimensional accuracies that concrete and timber buildings require, and their subsidence, also differ substantially [86].

Pan et al. [108] present a hybrid timber–concrete building composed of two parts: a concrete core with concrete flat slabs on every third floor as the main structure and prefabricated light timber-frame modules as substructures. Two out of three concrete slabs are replaced by timber modules to create the livable spaces. The concrete slabs offer fire separation at each third level. Bolted connections were designed to connect the main structure and the substructures.

Most timber buildings have reinforced concrete foundations and basements if present. The exception is using timber piles or foundations with bolted metal cap beams.

Concrete cores and podiums with additional steel elements incorporated on the inside or outside the structure are more common. Some projects have entire blocks made of concrete or steel, and balconies and exterior circulation areas are often constructed with these materials.

### 10.2. Module Level

In the context of structural solutions for prefabricated modules, it is essential to note that there are numerous intersections among them, regardless of the type of material used, whether timber, concrete, or steel.

Three generic forms of modular construction exist: continuously supported, four-sided, or panelized/monolithic modules, where vertical loads are transmitted through the walls; open-sided, corner-supported or framed at the edge's modules, where vertical loads are transmitted through corner and intermediate posts; and non-load bearing modules, often called pods, that are supported by the floor or a separate structure. These three construction methods are employed in diverse applications, contingent on the need for cellular spaces, like hotel bedrooms, or open-plan areas [91,93]:

- Continuously supported: Continuously supported or four-sided modules are supported on their longitudinal sides, which bear on the walls of the modules below. Continuous-supported modules are divided into a lightweight load-bearing wall system and a massive load-bearing wall system based on the type of sidewall support system [109]. Monolithic load-bearing wall modules are commonly used in concrete buildings, in which the concrete walls are used to transfer gravity loads to the foundation and resist the lateral loads [110]. The end walls are usually highly perforated with large windows at one end and a door and service riser at the other. In some systems, the edge beams in the floor and ceiling provide an indirect means of load transfer by bearing on each other, but in most cases, the side walls provide direct load transfer. The indirect transfer method relies on the resistance of the edge beams to compression across their depth, so this type of modular construction can be more limited in height. The direct transfer of loads through the side walls relies on the compression resistance of the structural elements. In continuously supported modules, corner posts can be incorporated to serve as lifting points and connection points to other modules and structural elements. Stability is provided either by placing X-bracing in the walls of the modules, by the diaphragm action of sheathing boards, or by a separate bracing system [83,93];

- Open-sided: Corner-supported modules have posts at their corners and sometimes at intermediate points, and edge beams span between them. In this way, the modules may be designed with open sides, although infill walls can be used to form the cellular space. Because of that, this solution usually has more flexibility in architectural design and larger module sizes. This leads to fewer modules and connections [91,96]. They are generally made of steel in which the gravity loads are transferred to the slab, the edge beams and corner columns, and the foundations [110]. Therefore, frame-supported modular units are employed in high-rise MSBs due to the robust compression, torsional, lateral-torsional, and bending resistance of columns and beams [111]. However, because the beam-to-post connections are relatively weak in terms of their bending resistance, the stability of the group of modules is provided by additional bracing often located around the stair and lift core [93]. The lateral loads due to winds and earthquakes are transferred to shear wall cores and bracing systems via ceilings and floors [93,111]. These types of open-sided modules are often used in the health and educational sectors [93]. The corner-supported module is commonly used in high-rise buildings due to its high load-carrying capacity [110]. Another variation is the open-ended modules that can be manufactured so full-height glazing can be provided, and modules can also be combined along their length [93];

- Non-load bearing: The third type is non-load bearing modules, which can only support their weight and crane lifting forces. Simultaneously, relying on adjacent members or modules, such as wall support landings and half-landings of staircase modules, resist externally exerted forces. On the other hand, the bathroom, kitchen, and plant-room modules are installed on finished floors to transfer external loads [112].

Timber framing has been a prevalent construction method in the residential sector for modular housing since the 1960s, especially in the United States. Initially used in modular construction, particularly in temporary or relocatable buildings, this approach relies on prefabricated timber wall panels utilizing wall studs with a corresponding top and bottom track. Typically sheathed with plywood or oriented strand board (OSB), these wall panels incorporate one or two layers of plasterboard on the interior. The external walls are insulated externally with rigid insulation boards or with mineral wool placed between the timber studs. There is also a vapor barrier and a waterproof membrane. Floor and ceiling panels, designed with deeper joists, may include edge beams such as deep laminated beams, enabling up to 10 m spans between corner posts [93]. To optimize the strength and stability of the structure, it is a common practice to use varied stud spacings depending on the loads to which the modules are subjected: modules on lower floors have reduced stud spacing, while modules on upper floors, with less load, have larger stud spacings.

Over time, timber modular systems have evolved into versatile solutions suitable for both temporary and permanent constructions, accommodating a variety of functions. The construction techniques employed in timber buildings encompass timber framing for low-rise timber structures and post-beam and paneling systems for high-rise mass-timber projects, both strategically organized according to modular principles [81]. However, systematically dividing timber construction into more general methods such as panel, frame, and solid timber construction no longer seems reasonable in this context. By intelligently combining various building elements, customized solutions can be created in practice, giving planners the greatest possible design freedom. One frequent combination is timber panel exterior walls and ceilings and load-bearing interior walls made of dowel laminated timber or glued laminated timber, using the elements' thermal insulation, soundproofing, and fire safety advantages. Planar structural components have always been combined to brace structures and enclose spaces in frame structures [86].

The widespread practice of combining timber with other materials, such as reinforced concrete or steel, in construction has led to the creation of timber–concrete composite (TCC) and steel–timber composite (STC) components [106].

TCC solutions are commonly applied to beams, floors, and walls at the module level. TCC beams consist of a timber beam coupled with reinforced concrete in the transversal or longitudinal direction. TCC slabs, a well-known hybrid component, feature a slab-type EWP or timber beams in a joisted system connected to a reinforced concrete slab through shear connectors. TCC walls are created by connecting a massive timber or framed structure to a reinforced concrete slab with various materials such as solid timber, GLT (glued laminated timber), LVL, adhesive-free EWPs, regular concrete, lightweight concrete, or high-performance concrete. TCC fabrication can occur in wet–dry or dry–dry construction systems, involving processes where wet concrete is poured onto dry timber or precast concrete is connected to the timber part, respectively [106].

STC solutions are frequently employed in construction, especially when there is a need to distribute heavy loads across multiple points. Steel elements are used as connectors in frame construction, integrated into box or slab elements, and inserted in prestressed beams or frame structures through steel cables. This integration allows for longer spans, flush beams and joists, and more slender support cross-sections in modern timber buildings. Additionally, when steel elements are incorporated into timber, the surrounding timber provides fire safety, eliminating the need for a separate fire protection coating for the steel to ensure a sufficient duration of fire resistance [86].

In certain situations, edge timber modules on both longitudinal sides of the building incorporate a combination of steel and wood to resist lateral movement. Steel frames can be fully concealed within the timber cavities to maintain a seamless appearance and prevent external observers from discerning the difference [81].

## 11. Stability

### 11.1. Building Level

In high-rise buildings, lateral loads from winds and earthquakes are counteracted by an independent lateral stability system, such as a bracing truss or shear wall core. The lateral stability system can be constructed from steel, concrete, and hybrid steel–concrete structures using either traditional or modular construction methods [104]. While timber modular structures are primarily confined to low-rise applications, steel and precast modular units have established themselves as the industry standard for high-rise projects [109].

In contrast to the bracing truss, the shear wall core has the flexibility to accommodate elevators, stairs, and service risers. In current practice, four out of five of the world's tallest modular buildings constructed by the 3D volumetric method are based on concrete shear wall cores. Concrete cores can be constructed using either conventional methods, such as slipform and jumpform, or through the modular approach employing precast concrete systems. The composite steel–concrete–steel sandwich system is another option for the shear wall core [83]. Due to the discontinuity in the structural components among modules, the modules are designed to resist the gravity loads only whereas the core wall resists all the lateral loads. The result is a laterally stable building if the inter-module connection is designed correctly to transfer the horizontal load via axial and shear forces. The floor slab design in each module needs to be stiff enough so that each can act as a rigid plate [113].

While the use of pure timber structures for medium- and high-rise buildings remains a challenge, it is undoubtedly feasible up to a certain height. The Mjøstårnet tower in Norway, standing at 18 stories, exemplifies current possibilities and challenges. Additional mass, such as screed or concrete slabs without composite action, has been introduced to control sway and vibration. Although visible, the stabilizing system's diagonals sometimes impose limitations on functionality [114].

A typical 50–60 m tall timber high-rise building usually consists of a concrete core (to meet fire protection, acoustic, and structural requirements) in combination with timber or timber–concrete composite floors and possibly timber columns along the facade (at relatively short intervals). For taller buildings, additional bracing for sway vibration and comfort requirements or additional stability systems are usually also required [115].

### 11.2. Module Level

To facilitate the transfer of lateral loads from the module to the lateral stability system, supplementary bracing systems should be incorporated within the floors and ceilings of the module. Also, the inter-module connections should have sufficient shear strengths [83].

Bracing is often achieved by double-sided cladding of the framework with panels made of a processed timber construction material to enhance stability. Alternatively, internal bracing with plasterboard is employed, ensuring structural integrity, and contributing to the overall strength of the module [116].

When additional stabilizing elements, such as wall cladding, are to be avoided, the structure can be braced with rigid corner elements. However, the total allowable height of these structures is limited to three stories due to fire protection and structural constraints. For buildings exceeding three stories, non-structural timber construction modules can be integrated into load-bearing steel or reinforced-concrete frame structures [116].

The complexity of a structure rises proportionally with the overall height of the building. For low-rise buildings (three to five stories), where the forces to be resisted are relatively low, shear wall systems mainly made of CLT panels can be used. Alternatively, timber-frame systems arranged around the perimeter can be provided by bracing. Moment-bearing structures (rigid frames) rely on the interaction between beams and columns that

transfer the moment caused by horizontal forces directly at the connection. Due to the typically low strength and stiffness of timber moment connections, this type of system is not very common [114].

## 12. Joining Techniques

The design of modular building systems is highly influenced by structural integrity against critical loading conditions. One of the main aspects is to ensure that all the assembled parts can, together, have a lateral load resistance, a compression capacity related to vertical forces and load-transfer, and a deformation capacity within the connections between the structural elements and modules.

In terms of connections, a modular building connection system can be divided into three different types: (i) inter-modular (IMC); (ii) intra-modular; and (iii) module to foundation (Figure 9).

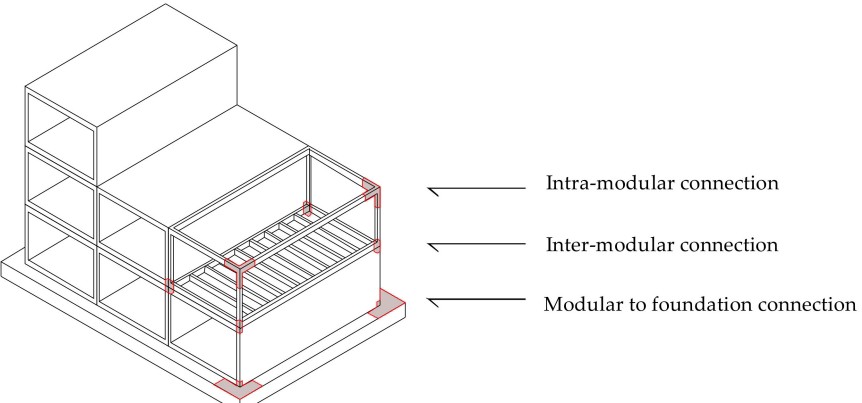

**Figure 9.** Types of modular building connections. Adapted from Rajanayagam et al. [117].

The inter-modular connections are crucial for determining the overall structural behavior of the system [118] since they are mainly responsible for the horizontal and vertical links within stacked modules. The inter-module connections between adjacent modules are pivotal for maintaining the integrity, stability, and robustness of modular building systems [119].

Intra-modular connections refer to those within a module, which are crucial for maintaining the integrity and stability of each individual module. These connections also play a vital role in ensuring that the module's structural behavior is properly supported and transferred to adjacent modules, all the way down to the foundation.

The module-to-foundation connection governs the sliding and overturning of entire modular building structures. An important aspect to consider in these connections is the fact that these failures, caused by lateral forces, are mostly due to how well attached the entire structure is to its base [117].

The Design for Manufacturing and Assembly (DfMA) principle has increasingly demanded that innovative connections be designed and tested so that performance can be directed and driven to fulfill requirements such as seismic design, robustness, and lateral load resistance.

### 12.1. Intra-Module

For the assembly of timber elements composed of panels and modules, screws and metal connectors are predominantly employed. Among the metal connectors available in the market, notable options include galvanized steel brackets, perforated plates, metal shoes, and extruded aluminum profiles. The array of metal connectors within the modules encompasses the following:

1. Self-drilling screws with partial thread for timber, wide-head;
2. Self-drilling screws with partial thread for timber, countersunk head;
3. Self-drilling screws with full thread for timber;
4. Ring-shank nails;
5. Hold-down angle plate;
6. Shear angle plate;
7. Perforated plates.

Numerous specifications are necessary for connectors to join elements, including type, size, quantity, and angle, which must be carefully designed on a case-by-case basis and can be consulted in catalogues from specialized suppliers.

Concerning the geometry of the union between elements, the method of joining walls often involves simple juxtaposition, unlike the junction between walls and floors. The latter may require different solutions depending on factors such as architecture, fire safety, aesthetics, and the function and location of the elements.

### 12.2. Inter-Module

In modular buildings, ensuring structural integrity, overall stability, and robustness relies heavily on the connections between modular units. The inter-modular connections entail horizontal connections in two plane directions from neighboring modules and a vertical connection within stacked modules.

The main aspect is to provide a path for load sharing and transmission between modular units which are the linking elements that allow stacked modules to transmit loads effectively to the base. Another great point is that they fulfil robustness requirements through load sharing by avoiding catastrophic incremental collapse due to local failure.

Differential movements between modules are also restricted by inter-module connections to avoid both serviceability failure (e.g., high drift ratio) and damage to the layers' protections coats.

In terms of seismic performance, despite the fact that the behavior of multiple stories of modular units is not adequately understood, the inter-module connections play a key role by dissipating the energy to maintain the structural behavior between the joined elements [119].

While welded connections offer rigidity, they are not preferred during on-site construction due to the need for highly skilled labor, substantial working space, and time-consuming post-weld inspections. Various inter-modular connection techniques have been developed for modular buildings, categorized into three types [83]:

- Rod based;
- Connector based;
- Bolt based.

### 12.3. Module-to-Foundation Connection

The module-to-foundation connection is another essential type for modular building systems and it becomes more critical as the height of the building rises. Typically, for a post-and-beam system these connections are usually made by using a corner fitting or embedded column. In addition, for timber–hybrid systems, hold-down connectors or anchor bolts are responsible for the foundation attachment to the frames or panels (e.g., CLT panelized system).

## 13. MEP Installations

There are several possible solutions regarding the location of the building's infrastructure. Pipes can run through the ceiling, walls, floor, or even the intersections of different planes. The study conducted by Monsberger et al. [120] highlights several crucial principles to consider when designing solutions involving piping in timber structures. A recommended approach is to concentrate pipes in specific and well-defined areas, facilitating inspections and access for maintenance. This organization not only simplifies the iden-

tification of damage to the timber structure in the case of leaks but also expedites the implementation of corrective measures. The strategic installation of maintenance doors is another important consideration, providing sporadic access to facilities and contributing to the ease of inspections. In inaccessible locations, it is advisable to take additional measures to protect pipes, such as including tapes or reinforcement tubes with larger diameters. However, it is important to note that this approach may increase the size of the cuts. Despite being underdeveloped, electrical humidity monitoring systems are also mentioned as a possible solution for such situations. The same author also mentions that an alternative could be using prefabricated cores as an autonomous module, which can be added to and adapted to the proposed solutions.

Incorporating "service cores" within structural 3D modules is a significant development in offsite construction. These prefabricated modules, equipped with ample structural capacity, include crucial components such as kitchens, bathrooms, mechanical–electrical shafts, stairs, and elevator shafts in housing projects. Upon on-site installation, these cores are strategically positioned to support slabs, partitions, and envelope panels, creating spacious and adaptable "served" areas like living rooms, dining rooms, and bedrooms [91].

Specialized pods designed for toilets and kitchens are commonly found in educational and healthcare buildings, commercial establishments, and hotels. Since the mid-1980s, large bathroom/toilet units have been utilized in city center commercial buildings [93].

Pods, as non-load-bearing modular units supported by the main building structure, play a crucial role in construction, especially in highly serviced areas. Typically measuring 2.00 m × 2.40 m in external dimensions, these units come in various materials. Specialized manufacturers produce pods in standard designs and can customize them for larger production runs. Some pods, designed without floors, aim to minimize construction depth and weight, addressing concerns about stepping into a bathroom module. Modules with open bases often feature ceilings that serve as the floor for the module above, a solution commonly used in prisons and secure accommodations. Innovations include under-floor heating systems with pipes embedded in the ceiling slab. Connection points on the exterior wall of these pods facilitate the integration of the main building's services and drainage systems [93]. An alternative could be prefabricating the utility core as an autonomous module, which can be added and adapted to the proposed solutions [120].

To optimize the utilization of shared service risers, bathroom pods are commonly positioned back-to-back around the service riser, allowing for the concentration of up to four pods in one area of the slab. While some situations may permit a step from the general floor level into the bathroom, in most cases, a thinner floor is necessary under the bathroom pod, or a layer of screed is applied over the adjacent floor to align it with the pod's floor level [93].

To enhance building design, there is a desire to reduce the construction depth and increase the headroom in modular structures. In contrast to conventional buildings, where a single beam supports both the ceiling of the lower story and the floor of the upper story, modular structures consist of separate units that are stacked. Each module has a ceiling beam supporting the MEP services and a floor beam supporting dead and imposed loads. When modules are stacked vertically, the two beams leave a small gap between the upper and lower stories, resulting in extra vertical space consumption compared to conventional buildings. If a more significant depth of the floor beam is required due to a long span or higher design-imposed load, the construction depth increases, further sacrificing headroom [96].

## 14. Design for Disassembly

The topic of flexibility in construction is not new. For Groák [121], the definition can be divided into two types of intervention. On the one hand, adaptability can be achieved by designing rooms or units that have many uses, mainly accomplished through arranging rooms, flow patterns, and space designations. On the other hand, flexibility is reflected in making changes to the building's physical structure, such as by expanding or adding rooms or units and utilizing sliding or folding walls and furnishings. Lehmann

and Kremmer [122] also point out the necessity of identifying that there are two phases: pre-configuration in the design phase and reconfiguration during the use of the building. This means recognizing that the hybrid nature of timber construction potentially allows for the creation of systems with different configurations, which can even be combined with other structural systems. However, there are many other factors besides the construction itself. According to Askar et al. [123], it is necessary to define the term adaptability as the ability of buildings to change in response to different needs, including social and local factors (available materials or user demands), environmental issues (climate change, environmental disasters), technical requirements (technological developments), economic factors, and bureaucratic issues (regulations), among others. However, there is a consensus about the numerous interpretations: true adaptability lies in the ability to respond to the impossibility of predicting future changes.

Design for Disassembly (DfD) is a theme frequently used over time by industries to acknowledge the need to maintain and repair their products. Its main objective is to reduce the consumption, cost, and waste of materials, eliminating waste that cannot be reintroduced into production cycles and increasing their life cycle. In the building industry, this can be carried out by simplifying systems and connections so that buildings function as stores of replaceable materials over time. According to Bogue [124], the connection points must be made up of reversible connections with good accessibility and visibility so that they can be dismantled using only mechanical fixings. Despite the variation in opinions on this theme, concepts like circularity, which includes maintenance and repair, adaptation, relocation, reconfiguration, and reuse, are all based on DfD principles [125] (Figure 10).

Developments around these concepts have triggered greater investments in reusing systems and components in construction. While recent years have seen a paradigm shift towards transforming linear production systems into circular ones, the current focus is on reducing the flow of this circularity to allow for greater longevity in construction.

Within the scope of academic research, there are several ways of categorizing the different types of adaptability that serve as proof of the consequent exploration of the topic; however, the multitude of terms reveals that there is a problem in standardizing these concepts. One of the most common classifications in academic circles, which seeks to standardize the various existing concepts into sub-categories, is that of Schmidt II and Austin [126].

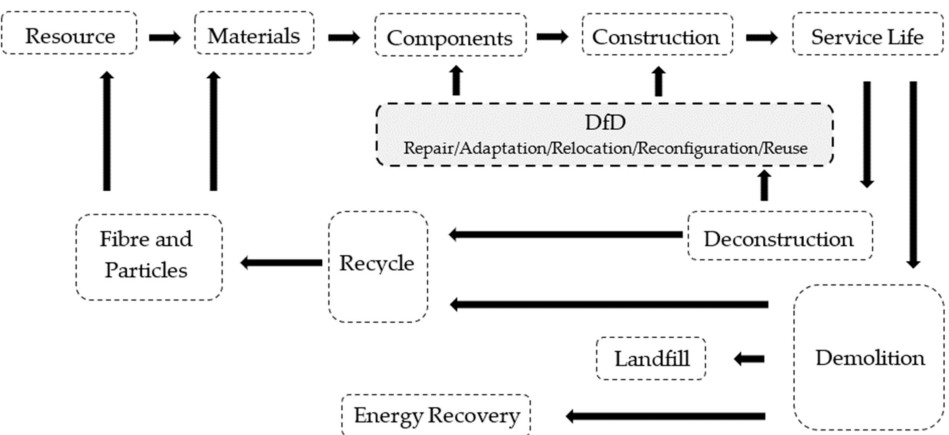

**Figure 10.** Reuse principles on timber life cycle. Adapted from Ottenhaus et al. [125].

The division was created to establish a standard for design models and their main fundamental characteristics to allow for adaptability in architecture, with a spectrum of options that allow for multiple changes over the lifetime of buildings [127]. In this sense, the proposed sub-categories are adjustable, versatile, refitable, convertible, scalable, and movable. Despite the unanimity of this categorization, there are other definitions that, despite grouping the same concepts and using them as a basis, try to systematize

the information differently. Magdziak [128] organizes the different types of adaptability based on location, scale, and intended function, while Schneider and Till [129] distinguish between different parameters, from spatiality to technical construction issues.

As pointed out previously, one of the main challenges in this type of system is the bonding between materials. Wood is a composite of structured natural fibers, meaning loads that exceed the material's maximum elastic limit will cause it to break [125]. Demands for reduced cost and speed of execution mean that most market solutions opt for using chemical connections to link components, giving rise to monolithic solid constructions, which essentially focus on building a final product [130]. Even though this type of solution is effective, it has several limitations regarding future disassembly.

According to Dams et al. [131], the use of dry or mechanical connections should be favored, along with the use of screws, bolts, or staples, which are easily accessible and uniform. This type of element should also be applied using gravity (without momentum), thus making the assembly and disassembly process easier. Stainless steel is one of the main materials for making durable and reusable connections. The main solutions employed can thus be categorized as dry solutions, which require a single material (wood), or those that use metal fittings, ranging from simple conventional solutions (simple fasteners) to more sophisticated, market-based, and highly standardized options. Although over the last few years dry solutions have been progressively replaced by metallic solutions, the technological advances in manufacturing processes (such as CNC), which allow for the precise construction of large-scale parts, the construction of systems based on traditional logics has been the focus of some recent research [125]. One example is the SI-Modular system developed by Metsa [132], which enables houses to be constructed completely from timber without screws by simply using interlocking connections in the installation. However, these developments are only isolated cases without many repercussions from the rigidity of the timber and the need to take care when exposing it to variations in temperature and humidity, which make it difficult for the connections to function, meaning that metal solutions continue to be the focus in the development of solutions. Some studies have conducted tests to understand the applicability of metallic solutions to different types of timber systems.

Yan et al. [133] applied CLT panel fixing solutions to timber-frame systems to assess their behavior. Some of the main conclusions were surrounding the deformations in the materials, which question their future disassembly and reuse, and they concluded that there is a need for future research on the subject. Also, Arvaniti [134] points out that one of the main challenges of disassembling timber systems is deformation (in the holes, fasteners, or the wood elements themselves), which alters their characteristics and presents lower resistance properties compared to new materials. According to Bogue [124], these deformations are inherent to the physical behavior of the material, and they should be a fundamental factor to take into account in the design phase, designing solutions in which the changes resulting from traction, bending, and corrosion, among others, should occur mainly in the replaceable metal elements.

Another major concern for companies in the sector is the compatibility of MEP infrastructures with reversible systems. Given the ever-increasing volume of infrastructure required to meet today's energy demands and living conditions, the possibility of moving the planes where this type of pipework is usually located is proving to be a challenge. According to Nelson [135], although the experience of locating installations on the floor makes it possible to redesign the non-structural elements inside the modules, installation on the walls seems to be the most widely used solution on the market.

Finally, one of the fundamental issues for a building's adaptability is the variations in scale and the consequences for the structure. As seen in previous sections, the structure of volumetric module systems is based on partition walls with a structural function. Removing or changing one of these walls requires the reinforcement of the entire system if this has not been analyzed and previously defined in the design phase [134]. In this sense, it is essential to make the structural solution compatible with any possible changes the building

may experience in the future. One example of the difficulty of making changes that have not been idealized in the design phase is the spacing of the substructure in lightweight systems, which conditions the future opening of doors between rooms. Since the metric used is a subdivision of the dimensions of the materials, ensuring that the smaller the spacing, the better the structural behavior of the wall, it is straightforward to use standard measurements of 60 cm spacing, which makes it impossible to open a future doorway. On the other hand, flexibility is facilitated in systems where the structure is separated from the rest of the building's components, as seen the study on the principles of Open Building and Shearing Layers, which provide total freedom in the definition of interior spaces since the dividing walls have no structural function.

All these challenges imply greater investment in the building design phase, which, according to Arvaniti [134], implies a significant increase in execution costs. However, according to a study conducted on a sample of 48 buildings in the US, the implementation of adaptability strategies resulted in only a 1% increase in the initial construction costs compared to conventional strategies [135].

## 15. Discussions

A comprehensive exploration of the current state of the art in modular timber construction has revealed a dynamic landscape marked by both notable achievements and inherent challenges. Our examination showed that structural design, transportation considerations, and material utilization underscore the evolving nature of this construction methodology. Insights from various studies shed light on the increasing interest in tall timber buildings, driven by the material's sustainability and unique structural properties. Despite advancements, the discussion also identified persistent hurdles, such as societal perceptions, regulatory hesitations, and gaps in knowledge. This overview sets the stage for a nuanced discussion of modular timber construction's development, challenges, opportunities, and knowledge gaps.

Timber, one of the oldest building materials and the dominant one in construction until the 20th century, saw its usage become limited due to public perception and restrictive regulations introduced for fire safety, contributing to the consolidation of the concrete industry in the market [63].

A decade ago, there was a notable absence of structural wood use in multi-story buildings globally, primarily limited to residential one- or two-story structures. However, a unique historical moment was also unfolding, suggesting that wood was transitioning from a material with a long history to one that would define construction for the next millennium. Significant potential awaits the expansion of modular timber constructions, especially within dense and sustainable urban landscapes.

Despite skepticism about the material's performance, stemming from years of its structural disuse and resistance to change due to our dependency on established standards, the recent resurgence of wood construction has primarily occurred through revisions to regulations, extra regulatory incentives from collaborative public–private sector entities promoting wood use, and cooperation between the construction and forestry industries and research institutes, aiming to enhance product refinement or even revive a preference for wood, making it the primary construction material of this century [136].

The construction of various timber buildings erected in recent years has showcased the multiple possibilities of wood and its technologies, restoring confidence in the material. This is reflected in the growing interest in such projects, whether motivated by ecological value, thermal qualities, structural attributes like advantageous bending behavior and a favorable weight-resistance ratio, or aesthetic appeal [29].

This was feasible mainly due to the introduction of Engineered Wood Products and the utilization of digital design and production through CAE (Computer-Aided Engineering), CAD (Computer-Aided Design), and CAM (Computer-Aided Manufacturing). Leading to innovative systems characterized by high quality and precision, these advancements

have streamlined and enhanced the prefabrication process for timber buildings and its associated benefits.

Many countries, however, still experience hesitancy regarding the safety and durability of wood in tall construction, resulting in few and conservative projects being executed or in progress [137]. Moreover, due to the recent emergence of modern wood construction techniques, particularly for tall and modular buildings, this typology has not been scientifically studied in depth or widely accepted [95]. This gap is evident in the collective understanding of the material and poses a significant research limitation. Despite consumers and industry professionals recognizing wood as a natural, renewable material with excellent thermal properties, it has not been universally regarded as strong, durable, modern, and fire-resistant, despite extensive advancements and discourse surrounding timber construction and its applications. Consequently, the wood industries remain fragmented, consistently confronting potential threats to their competitiveness and development, leading to a notable decrease in the dissemination of information on the subject. The divergence of opinion among industry professionals and consumers regarding skepticism and rejection of wood usage indicates a lack of training and understanding of the typology among various construction actors, such as engineers and architects. According to Roos et al. [138], architects acknowledge wood's environmental and aesthetic value but perceive it as culturally dependent. On the other hand, engineers appreciate its strength-to-weight ratio and new wood products but remain concerned about fire safety, sound transmission, implications due to moisture variations, and a shortage of specialized labor, preferring more familiar and explored materials. Additionally, authorities and contractors view wood structures as being synonymous with higher commercial risks. Despite consumers associating the material with environmentally friendly construction practices and considering it an important factor in decision making for product purchases and service hiring, they indicated a lack of knowledge about wood's contribution to reducing greenhouse gas emissions through carbon storage. In a study focused on the acceptance of LVL, McGregor et al. [139] highlighted that, while consumers recognized the sustainable benefit of choosing this product, which has a lower carbon footprint than steel, this factor did not significantly impact material selection.

It is conceivable that consumer perceptions regarding fire behavior, moisture, and biological agents result from instinctive motivations based on experience [19]. However, promotional campaigns for wood usage implemented, for example, in the United Kingdom and Austria have demonstrated that perceptions about the material can be altered among consumers and construction professionals [140]. Understanding markets and spreading knowledge about the negative perceptions associated with the typology are fundamental components for its progression, enabling the development and proposal of solutions capable of mitigating common misconceptions and consolidating the typology as an alternative construction method [19]. Social acceptance becomes vital for successfully implementing any technological innovation, especially those that are not competitively cost effective, such as wood structures compared to concrete and steel.

Beyond the challenge of social acceptance, many countries still exhibit a constrained market for wood construction, grounded in the lack of specific regulations and the scarcity of specialized professionals in the sector. Efforts to provide specific education for civil engineers and architects in the field of wood structures are rare and recent, contributing to a limited workforce [137]. Additionally, often only a fraction of native wood resources is utilized, with wood imports being prevalent for this typology, which represents a significant missed opportunity for economic, social, and environmental gains [141].

Furthermore, knowledge gaps are the primary obstacles to using structural wood [63]. The dissemination of tall timber buildings in countries with harsh climates has promoted the accumulation of knowledge and the proposal of suitable solutions for this specific scenario, which may not directly apply to other contexts. Therefore, there is a need to develop knowledge to adapt this typology to different situations, considering variables such as climate conditions favoring biological deterioration; temperature ranges; urban

regulations for new construction; national normative requirements for functional (thermal and acoustic) and fire safety; the diversity of available forests for supply and the species and products offered by the forestry sector; widely used construction materials in terms of quality and cost, whose application and maintenance are well understood by technicians and users; prefabrication technologies available to the wood construction sector; traffic regulations and limiting conditions for material transportation; and architectural and aesthetic preferences. This would enable the proposal of suitable, reliable, efficient, and durable construction systems that fully leverage the benefits of the material.

## 16. Conclusions

Within the domain of prefabrication and modularity in timber systems, a list of topics remains underexplored and warrants attention. For instance, further research about which solution (1D, 2D, or 3D) should be used in particular project scenarios is essential. Factors such as transport, the solution's applicability considering site issues, the flexibility of the resulting spaces, or the local producer's manufacturing conditions influence the choice made. An extended study of various case studies is necessary to standardize solutions and narrow the options according to the specific need.

Even in light of recent developments regarding a specific standard value in terms of module length and width, there is a need for a more in-depth studies that look at issues such as the different types of transport, the regulatory dimensions, and the less important issue of which layouts (in module or overlapping) allow for the best percentage of return on transportable space; the selection of the materials used, their direct relationship with the metrics and repercussions on the final solution; or even the requirements and dimensions of the different programs, taking into account the regulations governing the use of space.

There is also a need to develop BIM tools in the timber construction sector in the same way as they have been for the concrete and steel sectors, in which are quite advanced and even more interconnected. Much of the software used today is due to the ease with which parts can be prepared (including the design of fittings and holes) and direct communication with CAM manufacturing processes. However, there are some problems when it comes to cross-referencing with the main programs used in the construction sector, and it is necessary to analyze ways of interconnecting communication to facilitate the development of solutions in the future.

Contrary to current trends in concrete and steel buildings, timber structures show a prevalence of rectilinear, symmetric plans and regular, repetitive extrusions with a low degree of innovation, possibly due to the lack of development of solutions that allow for greater layout variability. In this sense, there is a need for further analysis and cataloguing of the different market solutions for connecting planes to promote future research that tries to develop versatile solutions that are reflected in different space configurations. This will also contribute to the modularization technique, balancing standardization and adaptability.

Within the domain of structural solutions, timber modular construction derives inspiration from the steel modular concept, incorporating its very linear member-based construction method, stability arrangements, and joint techniques (screws, welding, or adhesives). Nevertheless, steel modular construction holds a distinct advantage in technical advancement, boasting a more extensive array of solutions that enjoy greater diversity and public acceptance.

In terms of adaptability, despite the focus on modularization and prefabrication, most of the solutions still focus on the linear assembly process, focusing on developing a final product. This option makes it difficult not only to make changes to the period of use of the building but also to dismantle it at the end of its life cycle. In this way, it is important to explore research into the creation of new models that allow for the redesign, alteration, and dismantling of buildings, ensuring greater freedom both in the design phase and during the period of use of the building. Among the main challenges are developing reversible connections between the various components, implementing MEP, and ensuring a structural system that allows greater spatial freedom. It is also necessary to explore

the reuse of materials, given the progressive requirements for certification, which differ according to the regulations in each country. Reusing components could be challenging, and further research into the limitations of and protocols for its feasibility is essential (performance tests, certifications, among others).

Table 2 summarizes the main information about each topic covered in this state-of-the-art review.

**Table 2.** Overview of the main topics explored in this state-of-the-art review.

| Section | | State-of-the-Art Review |
|---|---|---|
| 3. Wood as a modern construction material for modular buildings | | • Initiative to promote wood building construction (France, Canada Japan, Sweden, or Finland);<br>• Collaboration between construction and forestry industries, research institutes, and governments;<br>• Positive marketing of wood use produced by incentive programs and collaborative research demonstration projects. |
| 4. Engineered Wood Products and prefabrication | | • Reduction in project timelines and costs (85%);<br>• Saving energy in the construction phase (44%);<br>• Performance during the use phase (by more than 7%);<br>• Avoid construction traffic, noise impact, disturbances to the local community, accidents, the need for traditional construction equipment, wet processes, waste, poor execution, improper material storage are eliminated and joints, gaps, and penetrations minimized;<br>• Lags significantly behind technological advancements observed for other construction materials. |
| 5. Principles of modularity in timber buildings | | • Escalate efforts to craft solutions that seamlessly integrate the supply chain from design to production;<br>• Establish a system wherein the components can be amalgamated and reconfigured to cater to diverse needs. |
| 6. Taxonomy of modular timber products | | • Panel and space module systems were the most common systems (2000–2010) predominantly employed in mid-rise projects (10 stories);<br>• From 2009, there is a significant increase in frame structures. |
| | 1D Linear Systems | • Five to eight-story buildings;<br>• Combinations of EWPs, such as CLT (cross-laminated timber) slabs, ribbed slabs, or composite floors. |
| | 2D Panelized Systems | • Medium-height (5–10 floors) and low-rise building projects (3–4 floors);<br>• Assembly process is simpler than traditional construction but more intricate than assembling 3D modules;<br>• Provide greater flexibility and more convenient transporting. |
| | 3D Modular Systems | • 6% to 10% exceed nine floors (max. nineteen floors);<br>• High level of repeatability and a significant ratio of wet to dry rooms;<br>• Most complete prefabrication (95%);<br>• Faster than the 2D approach (16%); |
| | Hybrid Systems | • Enhancing productivity for bathroom and providing maximum flexibility for other areas;<br>• Deliver both solutions becomes more intricate (coordination of the overall supply chain); |
| 7. Manufacturing, transportation, and installation dimensions | | • Typically, the cutting machinery utilized allows for dimensions of 3.50 m × 9.00 m;<br>• A standard truck, the panels are confined to the following specifications: Length: 13.50 m; width: 2.45–2.48 m; height: 2.50–3.00 m; load weight: 24.00–25.50 tons;<br>• Length of a typical modular unit: from 6 to 12 m (modules less than 5 m long: often functional or flexibly designed modules; modules longer than 10 m: generally made for apartments);<br>• Width of a typical modular unit: from 2.50 to 4.00 m (transportation conditions contribute to a range between 2 and 5 m);<br>• Height of a typical modular unit: from 2.80 to 3.50 m;<br>• The standard floor space for volumetric modules ranges from 15 to 25 m$^2$;<br>• The mass of timber modules ranges from 5 to 15 t, and the mass of equipment is less than 25 t. |
| 8. Layout design | | • Rectangular (alternative "L" shape);<br>• Two-dimensional system include most examples that deviate most from orthogonal shapes;<br>• Regular extrusions with a flat roof dominate all structural systems (1D/2D show more volumetric variations);<br>• One-dimensional element projects generally follow grid systems (dominant);<br>• Two-dimensional panel and 3D three-dimensional ordered by linear arrays. |

**Table 2.** *Cont.*

| Section | | State-of-the-Art Review |
|---|---|---|
| 9. Building occupancy and use | | • The most common is residential (half of all projects);<br>• Hotels, senior residences, student accommodations, hospitals, and schools have appeared more recently (repeatability);<br>• 1D—office buildings; 2D—residential buildings (whether for social housing, student housing, or multifamily housing); 3D—hotels and hostels. |
| 10. Structural systems for modular buildings | Building level | • Structural systems for tall buildings: rigid frame; core systems; shear wall systems; shear frame systems; mega core systems; mega column systems;<br>• Structural solutions to support volumetric modules: module stacking; module anchored to an exoskeleton; module anchored to the core;<br>• Structural components often made with concrete/steel when in timber-hybrid buildings: podium or plinth; core; floor slab; lateral bracing; external structural elements (such as stairs and circulation areas); |
| | Module level | • Generic forms of modular construction: Continuously supported (or four-sided modules); Open-sided (or corner-supported or framed at the edge's modules); Non-load bearing modules (often called pods);<br>• TCC solutions: beams; floors; walls;<br>• STC components: connectors; box or slab elements; cables; |
| 11. Stability | Building level | • Four out of five of the world's tallest modular buildings constructed using the 3D volumetric method are based on concrete shear wall cores;<br>• A typical 50–60 m tall timber high-rise building usually consists of a concrete core in combination with timber or timber-concrete composite floors; |
| | Module level | • Bracing is often achieved by double-sided cladding of the framework with panels made of processed timber construction material. Alternatively, internal bracing with plasterboard is employed;<br>• When additional stabilizing elements, such as wall cladding, are to be avoided, the structure can be braced with rigid corner elements; |
| 12. Joining techniques | Intra-module | • Self-drilling screws with partial thread for timber, wide-head or countersunk head; self-drilling screws with full thread for timber; ring-shank nails; hold-down angle plate; shear angle plate; perforated plates. |
| | Inter-module | • Rod-based; connector-based; bolt-based. |
| | Module to foundation | • Corner fitting or embedded column (1D); Hold-down connectors or anchor bolts (hybrid systems). |
| 13. MEP (mechanical, electrical, and plumbing) installations | | • A recommended approach is to concentrate pipes in specific and well-defined areas, facilitating inspections and access for maintenance;<br>• Incorporating "service cores" within structural 3D modules;<br>• Specialized pods (2.00 m × 2.40 m) are designed for toilets and kitchens (educational, healthcare, commercial and hotels. |
| 14. Design for disassembly | | • There is a need for future research on the fixing solutions between different systems;<br>• Concern in the compatibility of MEP with reversible systems;<br>• 1% increase in initial construction costs compared to conventional strategies. |

**Author Contributions:** Conceptualization, M.T. and J.M.B.; methodology, M.T.; data curation, M.T., R.F. and V.B.; writing—original draft preparation, M.T., R.F., V.B., F.S., C.M. (Cláudio Meireis), M.F., I.V., A.G. and R.A.; writing—review and editing, M.T., R.F., V.B., F.S., C.M. (Cláudio Meireis), M.F., I.V., A.G., R.A., S.M.S. and J.M.B.; supervision, S.M.S., D.L., A.F., C.M. (Carlos Maia), A.C. and J.M.B. All authors have read and agreed to the published version of the manuscript.

**Funding:** This research was supported by the doctoral grant PRT/BD/152841/2021 financed by the Portuguese Foundation for Science and Technology (FCT), with funds from the State Budget and the community budget through the European Social Fund (ESF), under MIT Portugal Program; and by the R&D Project "R2U Technologies | modular systems", with reference C644876810-00000019, funded by PRR—Plano de Recuperação e Resiliência—and by the European Funds Next Generation EU, under the incentive system "Agendas para a Inovação Empresarial". This work was partly

financed by FCT/MCTES through national funds (PIDDAC) under the R&D Unit Institute for Sustainability and Innovation in Structural Engineering (ISISE), under reference UIDB/04029/2020 (doi.org/10.54499/UIDB/04029/2020), and under the Associate Laboratory Advanced Production and Intelligent Systems ARISE under reference LA/P/0112/2020.

**Conflicts of Interest:** The authors declare no conflicts of interest.

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
