# Peer review of "Contemporary Strategies for the Structural Design of Multi-Story Modular Timber Buildings: A Comprehensive Review"

_applsci, doi:10.3390/app14083194_

Round 1

Reviewer 1 Report

Comments and Suggestions for Authors

This survey gives a comprehensive review of various aspects on for Multi-Story modular-timber buildings, based on a careful study on existing projects and literature. This would be a great reference for audiences interested in this field, offering an entire landscape.

Section 5 Principles of modularity in timber buildings has a strong favor on prefabrication in factory. But how about making the site as a temporary factory (as recent 3D printing projects)? How about using intelligent robots do all work in site? See this robotic excavation example: Johns, R. L., Wermelinger, M., Mascaro, R., Jud, D., Hurkxkens, I., Vasey, L., ... & Hutter, M. (2023). A framework for robotic excavation and dry stone construction using on-site materials. Science Robotics, 8(84), eabp9758.

There has been innovation in swarm assembly or self-assembly robotics:

Leder, S., Weber, R., Wood, D., Bucklin, O., & Menges, A. (2019, June). Design and prototyping of a single axis, building material integrated, distributed robotic assembly system. In 2019 IEEE 4th International Workshops on Foundations and Applications of Self* Systems (FAS* W) (pp. 211-212). IEEE.

There is a large in-between area between strict modularity and freeform. The discrete design in architecture explored this topic in depth:

Sanchez, J., 2019. Architecture for the Commons: Participatory Systems in the Age of Platforms. Architectural Design, 89(2), pp.22-29.

Retsin, G., 2019. Bits and Pieces: Digital Assemblies: From Craft to Automation. Architectural Design, 89(2), pp.38-45.

Section.8 discusses layout problem and figure.5 shows examples of structural ordering systems. Besides, simple grid or parallel system, there is a more systematic treatment on 3D grid about symmetry, see:

Hua, H., Hovestadt, L., & Li, B. (2022). Reconfigurable Modular System of Prefabricated Timber Grids. Computer-Aided Design146, 103230.

As a journal paper, containing 37 pages, it still seems too ambiguous to cover the topic. I am not sure how the journal treats such a long paper. From the viewpoint of research, there is no clear separation between problem and solution throughout the paper. For example, in 10. Structural systems for modular buildings, the solutions are mixed with the problem, which does not help the readers to reason about the key problems.

The paper seems to presume that timer buildings are environmentally friendly (though I believe so), but this needs to be seriously verified by data and analysis, in contrast to popular talk in social media or newspapers.

As the paper title emphasized ‘Innovative’, it ignored the ‘digital fabrication’ and ‘robotic + AI’ movement in architecture, for example:

Willmann, J., Knauss, M., Bonwetsch, T., Apolinarska, A. A., Gramazio, F., & Kohler, M. (2016). Robotic timber construction—Expanding additive fabrication to new dimensions. Automation in construction, 61, 16-23.

Wagner, H. J., Alvarez, M., Kyjanek, O., Bhiri, Z., Buck, M., & Menges, A. (2020). Flexible and transportable robotic timber construction platform–TIM. Automation in Construction, 120, 103400.

Kunic, A., Naboni, R., Kramberger, A., & Schlette, C. (2021). Design and assembly automation of the Robotic Reversible Timber Beam. Automation in Construction, 123, 103531.

Thoma, A., Adel, A., Helmreich, M., Wehrle, T., Gramazio, F., & Kohler, M. (2019). Robotic fabrication of bespoke timber frame modules. In Robotic Fabrication in Architecture, Art and Design 2018: Foreword by Sigrid Brell-Çokcan and Johannes Braumann, Association for Robots in Architecture (pp. 447-458). Springer International Publishing.

Author Response

R1: This survey gives a comprehensive review of various aspects on for Multi-Story modular-timber buildings, based on a careful study on existing projects and literature. This would be a great reference for audiences interested in this field, offering an entire landscape.

A: We would like to express our sincere gratitude to you, the reviewer, for your invaluable contribution to the quality of this work. Your insights and suggestions have significantly enhanced the content and clarity of the article. The time devoted to the detailed analysis and your dedication to providing a thorough and constructive review are truly appreciated.

R1: Section 5 Principles of modularity in timber buildings has a strong favor on prefabrication in factory. But how about making the site as a temporary factory (as recent 3D printing projects)? How about using intelligent robots do all work in site? See this robotic excavation example: Johns, R. L., Wermelinger, M., Mascaro, R., Jud, D., Hurkxkens, I., Vasey, L., ... & Hutter, M. (2023). A framework for robotic excavation and dry-stone construction using on-site materials. Science Robotics, 8(84), eabp9758.

A: The perspective of utilizing the site as a temporary factory has proved to be exceptionally insightful and valuable for our article, particularly in the context of a material whose sustainable appeal significantly drives its modern usage, as is the case with timber. This approach not only proves efficient in terms of production but, importantly, it ensures that the ecological value associated with timber usage remains intact, as it mitigates concerns arising from long-distance transportation inherent in commercialization of modern timber products.

The following passage was added to the article between lines 315-332:

Automated building processes facilitating efficient in situ resource utilization hold the potential to streamline construction operations in remote areas while offering a carbon-reducing alternative to conventional building practices [72] by minimizing transport distances, costs, and associated pollutant emissions. Importantly, it ensures that the ecological value associated with timber usage remains intact, as it mitigates concerns arising from long-distance transportation. The proximity between the factory and the assembly site enhances operational efficiency and streamlines quality control processes by enabling direct supervision and communication throughout the production cycle. Moreover, this strategy fosters prompt responses to project changes, offering greater flexibility and agility in tailoring components to meet specific project requirements. Unforeseen alterations in the project can be accommodated without significant delays in component delivery, while reduced waiting times for prefabricated components translate to increased on-site productivity. Furthermore, this solution stimulates local economies through job creation or the support of local sawmills and can benefit from the use of native materials. This approach can be partially implemented, focusing solely on essential services completed at the on-site facility. For instance, assembling and equipping prefabricated panels into modules on-site helps circumvent logistical challenges and economic concerns associated with transporting fully assembled 3D modules.

R1: There has been innovation in swarm assembly or self-assembly robotics:

Leder, S., Weber, R., Wood, D., Bucklin, O., & Menges, A. (2019, June). Design and prototyping of a single axis, building material integrated, distributed robotic assembly system. In 2019 IEEE 4th International Workshops on Foundations and Applications of Self* Systems (FAS* W) (pp. 211-212). IEEE.

A: Currently, factories devoted to prefabricating timber constructions are highly automated and consistently seek technological resources to optimize production while ensuring quality and precision in assembly. The perspective on robotics in handling and assembling components addresses a pertinent topic that indeed reflects the current (or desired) reality of present-day manufacturing facilities, which was the primary objective of this article.

With regards to this subject, the following passage was added to the article between lines 337-349:

During the modeling phase, meticulous attention is given to defining the provisions and dimensions of each element and its components, ensuring compliance with structural requirements while respecting dimensional limitations identified in design, production, transportation, and on-site application. Employing material optimization strategies is imperative to minimize cuts, thereby reducing production times and waste. This preparatory stage is pivotal for subsequent phases, requiring thorough planning to effectively manage technical and process performance. Following this, the CAD/CAM platform automates the generation of detailed instructions for CNC equipment, facilitating precise cutting and drilling of elements. Further phases, including lifting, handling, positioning, assembling, and connecting the diverse components, are seamlessly executed by automated systems. This holistic integration optimizes construction processes, bolstering efficiency, consistency, accuracy, and safety, consequently streamlining workflows, and enhancing project outcomes.

R1: There is a large in-between area between strict modularity and freeform. The discrete design in architecture explored this topic in depth:

Sanchez, J., 2019. Architecture for the Commons: Participatory Systems in the Age of Platforms. Architectural Design, 89(2), pp.22-29.

Retsin, G., 2019. Bits and Pieces: Digital Assemblies: From Craft to Automation. Architectural Design, 89(2), pp.38-45.

A: Indeed, it is crucial to incorporate these two concepts into the article, especially because the aim is for it to serve as a tool in the pursuit and development of increasingly versatile modular systems. These systems should accommodate the changes and adaptations necessary for designers to maintain creative freedom and for users to have a range of options and a versatile environment that meets their needs over time. The inclusion of these concepts in the text has contributed to substantiating and introducing the concept of a “kit of parts”, which was already present in the original version of the article.

The following paragraphs were added to the text between lines 372-386:

While in the freeform construction buildings are designed with little or no adherence to regular geometrical shapes or standardized components, allowing for highly customized and innovative architectural designs but demanding more time and resources, by adopting the modularization recent adaptative modular construction technique, designers and builders seek to balance standardization and adaptability, aiming for an efficient and versatile system. This approach aligns with broader trends in modular construction and prefabrication, emphasizing the benefits of standardized, repeatable elements in the built environment, such as quality, speed of construction, and cost-effectiveness.

The discrete design agenda has played a pivotal role in transitioning from strict modularity to adaptive modular construction. In this approach, fully functional and intricate buildings are assembled from serially repeating, rearrangeable sets of generic discrete elements, each treated as an individual entity with clear boundaries [86]. This approach diverges from the serialized production of identical units or generic solutions; instead, it relies on the combination and variation of purposefully designed parts to achieve customization and adaptability through scalable principles [87].

R1: Section.8 discusses layout problem and figure.5 shows examples of structural ordering systems. Besides, simple grid or parallel system, there is a more systematic treatment on 3D grid about symmetry, see:

Hua, H., Hovestadt, L., & Li, B. (2022). Reconfigurable Modular System of Prefabricated Timber Grids. Computer-Aided Design146, 103230.

A: The article's approach to symmetry in the 3D grid is highly intriguing and significantly complements the analysis of the 2D grid presented earlier. Considering symmetry in both dimensions provides a more comprehensive and in-depth understanding of the modular structure in question. The following excerpt has been added to the text (lines 751-754) so that interested readers can also refer to the cited bibliography.

According to Hua et al. [100], the symmetry in a grid is essential to the modularity in design and construction. Multiples are the choices around the symmetry in an architectural floor plan, but these decisions determine different prefabrication and logistics conditions.

R1: As a journal paper, containing 37 pages, it still seems too ambiguous to cover the topic. I am not sure how the journal treats such a long paper. From the viewpoint of research, there is no clear separation between problem and solution throughout the paper. For example, in 10. Structural systems for modular buildings, the solutions are mixed with the problem, which does not help the readers to reason about the key problems.

A: The authors acknowledge that the manuscript is indeed quite lengthy. However, this length was necessary to compile all the information deemed relevant for a state-of-the-art review on the topic. Despite its extensive nature, the reviewer identified gaps with interesting and pertinent topics that could be added. The length of the article led to a more sequential approach, where problems are identified, and solutions are presented immediately afterward. This approach helped prevent the text from becoming repetitive and even longer.

R1: The paper seems to presume that timer buildings are environmentally friendly (though I believe so), but this needs to be seriously verified by data and analysis, in contrast to popular talk in social media or newspapers.

A: A widespread understanding of the ecological value of timber constructions is one of the key pillars in combating prejudice against the use of wood as a building material. The following paragraphs have been added to the introduction of the article (lines 45-59) to provide information about the renewable, biodegradable, and carbon-storing nature of wood, and to contribute to the overall understanding of the ecological value of timber construction so that, based on informed knowledge, one consistently adheres to this principle.

The alarming global climate scenario, which has rapidly worsened in recent decades, has highlighted the urgency of reversing consumption patterns practiced by various economic sectors, demanding the establishment of increasingly ambitious collective climate and energy mitigation goals. As a growth strategy for a modern, competitive, resource-efficient economy, and consequently, conducive to sustainable development, one of the priorities outlined by the European Union is achieving carbon neutrality by 2050 [1].

In this context, the prioritization of revising traditional models replicated by the construction sector becomes crucial when confronted with data regarding their impact. The production and processing of materials for the sector hold the most significant share of energy consumption and greenhouse gas emissions [2], with cement and steel accounting for 4 to 7% [3] and 5% [4] of global CO2 emissions per year, respectively.

Wood, then, emerges as a promising construction material as an alternative to concrete and steel, with the benefit of its inherent contribution to emission reduction through its carbon storage capacity—approximately 0.9 t CO2/m³ of material—and lower embodied energy [5].

R1: As the paper title emphasized ‘Innovative’, it ignored the ‘digital fabrication’ and ‘robotic + AI’ movement in architecture, for example:

Willmann, J., Knauss, M., Bonwetsch, T., Apolinarska, A. A., Gramazio, F., & Kohler, M. (2016). Robotic timber construction—Expanding additive fabrication to new dimensions. Automation in construction, 61, 16-23.

Wagner, H. J., Alvarez, M., Kyjanek, O., Bhiri, Z., Buck, M., & Menges, A. (2020). Flexible and transportable robotic timber construction platform–TIM. Automation in Construction, 120, 103400.

Kunic, A., Naboni, R., Kramberger, A., & Schlette, C. (2021). Design and assembly automation of the Robotic Reversible Timber Beam. Automation in Construction, 123, 103531.

Thoma, A., Adel, A., Helmreich, M., Wehrle, T., Gramazio, F., & Kohler, M. (2019). Robotic fabrication of bespoke timber frame modules. In Robotic Fabrication in Architecture, Art and Design 2018: Foreword by Sigrid Brell-Çokcan and Johannes Braumann, Association for Robots in Architecture (pp. 447-458). Springer International Publishing.

A: The initial objective of the article was to gather as much information as possible about the structural and spatial strategies currently most employed in modular timber construction, especially those based on 2D and 3D solutions. Perhaps the choice of the word "innovative" for the title was misguided, as the article focuses on identifying established trends rather than those in recent ascendancy. Therefore, we suggest a title change to ensure that the article does not disappoint reader expectations:

Contemporary Strategies in Structural Design for Multi-Story Modular-Timber Buildings: A Comprehensive Review

Reviewer 2 Report

Comments and Suggestions for Authors

In their study, the authors explored the design approaches of contemporary multi-story modular timber buildings. A comprehensive literature review was performed to analyze wood as a construction material, engineered wood products and prefabrication, principles of modularity, taxonomy of modular timber products, manufacturing, transportation, and installation dimensions, layout design, building occupancy and use, structural systems for modular buildings, stability, joining techniques, mechanical, electrical, and plumbing installations, and design solutions for disassembly.

The selected topic is original and relevant in the construction field; it addresses specific knowledge gaps in modular timber construction. Research results contribute to the subject area and provide further knowledge about the design and construction of modular timber buildings, which are gaining popularity in recent years. The literature review is well structured; the results are appropriately discussed and illustrated by figures and tables. The list of references is extensive enough; it covers relevant state-of-the-art literature.

The manuscript could be interesting to the readers; however, minor revision is recommended before its publication:

1. Abstract (line 27). The abbreviation MEP should be avoided or explained in an abstract.

2. Methodology. The authors could provide a flowchart describing the main steps of the research. In addition, the authors could explain why they selected Google Scholar and Google Books engines to search for literature resources.

3. Section 3 (line 195). At present, HoHo Wien is not the “world's tallest wood building”. Taller timber buildings are Mjøstårnet in Norway and the Ascent MKE building in Milwaukee, USA. Please update the information.

4. Page 21, line 961. An abbreviation GLT shall be explained.

5. It is recommended to separate discussion and conclusions. In addition, research limitations have to be highlighted in the Discussion section.

6. The style of references does not fully correspond to journal requirements.

Author Response

R2: In their study, the authors explored the design approaches of contemporary multi-story modular timber buildings. A comprehensive literature review was performed to analyze wood as a construction material, engineered wood products and prefabrication, principles of modularity, taxonomy of modular timber products, manufacturing, transportation, and installation dimensions, layout design, building occupancy and use, structural systems for modular buildings, stability, joining techniques, mechanical, electrical, and plumbing installations, and design solutions for disassembly.

The selected topic is original and relevant in the construction field; it addresses specific knowledge gaps in modular timber construction. Research results contribute to the subject area and provide further knowledge about the design and construction of modular timber buildings, which are gaining popularity in recent years. The literature review is well structured; the results are appropriately discussed and illustrated by figures and tables. The list of references is extensive enough; it covers relevant state-of-the-art literature.

A: We would like to express our sincere gratitude to you, the reviewer, for your thoughtful review and invaluable feedback on our article. Your comments have greatly contributed to enhancing the quality of our work. We are delighted to hear that you found the selected topic original and relevant, and that our research addresses specific knowledge gaps in the field of modular timber construction.

R2: The manuscript could be interesting to the readers; however, minor revision is recommended before its publication:

Abstract (line 27). The abbreviation MEP should be avoided or explained in an abstract.

A: The meaning of the acronym MEP has been incorporated into the abstract at lines 27-28.

Moreover, inter-module joining techniques, MEP (mechanical, electrical, and plumbing) integration and design for disassembly are scrutinized.

R2: Methodology. The authors could provide a flowchart describing the main steps of the research. In addition, the authors could explain why they selected Google Scholar and Google Books engines to search for literature resources.

A: Indeed, even within a sequential process such as the methodology employed, a flowchart proves invaluable in facilitating clear comprehension and identification of the primary activities undertaken to accomplish the work. We appreciate the suggestion and have incorporated Figure 1 into the text at line 144.

Regarding the selection of Google Scholar and Google Books engines for literature resource search, the following excerpt was added to the text between lines 115-118:

These platforms provide a wide range of scholarly literature across disciplines, offering free accessibility and user-friendly interfaces for efficient navigation. Additionally, the inclusion of citation metrics assists in evaluating the impact of works, facilitating the identification of influential papers and authors.

R2: Section 3 (line 195). At present, HoHo Wien is not the “world's tallest wood building”. Taller timber buildings are Mjøstårnet in Norway and the Ascent MKE building in Milwaukee, USA. Please update the information.

A: We apologize for the significant error and appreciate your careful review. The excerpt containing outdated information has been removed from the text (line 218).

R2:  Page 21, line 961. An abbreviation GLT shall be explained.

A: The meaning of the acronym GLT has been incorporated into the text at lines 1026-1027.

TCC walls are created by connecting a massive timber or framed structure to a reinforced concrete slab with various materials such as solid timber, GLT (glued-laminated timber), LVL, adhesive-free EWPs, regular concrete, lightweight concrete, or high-performance concrete.

R2: It is recommended to separate discussion and conclusions. In addition, research limitations have to be highlighted in the Discussion section.

A: The separation of the discussion and conclusion sections was appropriately executed and significantly enhanced clarity and comprehension of the content. The following excerpt was added to the discussion section (lines 1384-1396) to emphasize the main research limitations identified in the process of compiling this state-of-the-art review.

Many countries, however, still experience hesitancy regarding the safety and durability of wood in tall construction, resulting in few and conservative projects being executed or in progress [138]. Moreover, due to the recent emergence of modern wood construction techniques, particularly for tall and modular buildings, this typology has not been in-depth scientifically studied or widely accepted [95]. This gap is evident in the collective understanding of the material and poses a significant research limitation. Despite consumers and industry professionals recognizing wood as a natural, renewable material with excellent thermal properties, it has not been universally regarded as strong, durable, modern, and fire-resistant, despite extensive advancements and discourse surrounding timber construction and its applications. Consequently, the wood industries remain fragmented, consistently confronting potential threats to their competitiveness and development, leading to a notable decrease in the dissemination of information on the subject.

R2: The style of references does not fully correspond to journal requirements.

A: Thank you for your careful review. We acknowledge the oversight regarding the style of references not fully corresponding to journal requirements. The format of the references has been revised, and the updated version can be found in the current file of the article.

Reviewer 3 Report

Comments and Suggestions for Authors

The research manuscript has presented a comprehensive overview of the present state of modular timber construction, highlighting critical areas that require further investigation. Additionally, the authors have put efforts in addressing production processes and design strategies that can positively impact the efficiency and sustainability of modular buildings. 

I would like to accept the manuscript in the present form for publication.

Author Response

R3: The research manuscript has presented a comprehensive overview of the present state of modular timber construction, highlighting critical areas that require further investigation. Additionally, the authors have put efforts in addressing production processes and design strategies that can positively impact the efficiency and sustainability of modular buildings.

I would like to accept the manuscript in the present form for publication.

A: We extend our sincere gratitude for your thorough review and feedback on our research manuscript. Your recognition of the quality and relevance of our work is immensely valuable to us, and we are truly honored by your endorsement. We appreciate your support, and we are committed to making further meaningful contributions to the field.